# Temporal dedifferentiation of neural states with age during naturalistic viewing
Selma Lugtmeijer [1] ✉, Djamari Oetringer[2], Linda Geerligs[2,4] & Karen L. Campbell [3,4]

While life is experienced continuously, we perceive it as a series of events. At a neural level, this event segmentation process has been linked to changes in neural states. An open question is whether neural states differ with age. Participants ($N = 577$) from the CamCAN cohort viewed an 8-min movie during functional magnetic resonance imaging. A data-driven state segmentation method was used to identify neural state changes. To study the effects of age, participants were sorted into 34 age groups. We show that neural states become significantly longer with increasing age, particularly in visual and ventromedial prefrontal cortices. Event boundaries overlapped with state changes in superior temporal and dorsomedial prefrontal regions, but there was no effect of age on this relationship. Our results suggest reduced temporal differentiation of successive neural states with increasing age. Nevertheless, preserved alignment between neural states and perceived events suggests coarse event segmentation remains intact.

According to Event Segmentation Theory[1,2], as we navigate through the world, we form representations of what is happening around us or 'event models'. These models help us to predict what will happen next. When our predictions fail, we perceive an event boundary and update our event model to better reflect what is happening in the next event. Event segmentation has often been studied at a behavioural level using movies and asking participants to press a button when they feel that one meaningful event ends and another starts[3]. Event boundaries identified with this procedure have been found to be reliable across participants and within participants over time[4]. Multiple studies have shown that more normative segmentation of events (i.e., being in high agreement with where others place event boundaries) is associated with better subsequent memory[5,6]. The identification of events depends on processing of perceptual features like movement, and processing of conceptual features like goals of actors[7]. Goals can consist of multiple subgoals or activities leading to a hierarchical structure of event boundaries in which coarse-grained events consist of multiple finer-grained events[8]. By instructing participants to attend to different temporal grains, they are able to identify both fine-grained and coarse-grained boundaries for the same activity[9].

With advancing age, there are differences in how we segment and memorize events. Some studies have shown that there is a reduction in segmentation agreement with age[9,10]. And that this affects how well these events are subsequently remembered. However, the evidence is mixed. Memory for naturalistic stimuli like movies tends to be relatively preserved with age compared to traditional lab-based tasks, such as learning lists of word pairs. And some studies have found similar levels of segmentation agreement with age[6]. The mixed evidence may be partially due to the fact that actively detecting event boundaries is effectively a double task, where participants have to attend to the story while also attending to potential boundaries. Indeed, implicit measures of event segmentation based on decreased reading speed at event boundaries or changes in eye-movements suggest that segmentation is preserved with age[11–15]. In this study, we aim to investigate this further by looking at segmentation at the neural level.

A number of studies have used naturalistic stimuli (such as movies) to examine the neural underpinnings of event segmentation. Despite the complexity of such stimuli, patterns of neural activation tend to show similarities across people[16,17]. When passively viewing a movie, different neural signals align with perceived event boundaries (as identified by either independent viewers or upon rewatching the film after imaging). Cortical areas that show an increased neural response around event boundaries are found in a wide network in the medial and lateral posterior cortex and some smaller frontal regions, including MT/V5, the posterior temporal sulcus, and frontal eye fields[18–20]. Several studies also suggest that the hippocampus is associated with event segmentation[18,21,22]. Hippocampal activity increases leading up to event boundaries with the magnitude being modulated by boundary salience[21]. Furthermore, more distinct activation patterns in the hippocampus between two adjacent events has been associated with better subsequent memory[22].

Recently, a novel data-driven approach has been introduced to study the neural underpinning of event segmentation[23]. Baldassano et al.[23] that

[1]Centre for Human Brain Health, University of Birmingham, Birmingham, UK. [2]Donders Institute for Brain, Cognition and Behaviour, Radboud University Nijmegen, Nijmegen, Netherlands. [3]Department of Psychology, Brock University, St. Catharines, ON, Canada. [4]These authors contributed equally: Linda Geerligs, Karen L. Campbell. ✉e-mail: Selmalugtmeijer89@hotmail.com

transitions between relatively stable brain activity patterns in different regions of the brain are organized into a nested structure with short neural states in sensory regions and long states in higher-order regions. This hierarchy has been observed across the entire cortex, and an overlap between neural state boundaries and perceived event boundaries was found throughout the entire hierarchy[24]. This overlap is particularly strong for state boundaries that are shared between multiple cortical regions[24]. Interestingly, it has been shown that individuals with more similar neural state boundaries have more similar memories of events in a movie[25]. In the current paper, we aimed to study event segmentation in an implicit way, by using this data-driven approach to study how segmentation of information into discrete neural states differs across the adult lifespan.

To this end, we used fMRI data from the full Cambridge Centre for Ageing and Neuroscience (Cam-CAN) adult lifespan sample ($N = 577$, age range 18–88) collected while participants viewed an 8-min movie[26,27]. This provides the opportunity to investigate if the temporal cortical hierarchy of neural states as reported in Geerligs et al.[24] is preserved with age. Secondly, examining age differences in overlap between perceived event boundaries and neural state boundaries might contribute to a better understanding of event segmentation in older adults and the neural mechanisms underlying conflicting findings in previous studies. In line with Geerligs et al., we used a data-driven state segmentation method (Greedy State Boundary Search, GSBS[28]) to identify neural state boundaries across the entire cortex in 34 age groups spanning the adult lifespan. Perceived event boundaries were based on an independent sample[21]. Based on prior findings of spatial dedifferentiation in the aging brain, which suggest reduced distinctiveness of neural representations for different objects and categories[29,30], we hypothesized that aging would be associated with less distinct neural states over time. Additionally, drawing on evidence of reduced suppression of previously attended information in older adults[31–34], we expected aging to be associated with longer-lasting neural states. This work provides insight into event segmentation across the adult lifespan, showing that while neural states become longer with age, their alignment with perceived events remains intact.

## Results

### Neural state duration hierarchy is preserved across the lifespan

To identify neural state boundaries, we applied Greedy State Boundary Search (GSBS[28]) to the 8 min fMRI data of an Alfred Hitchcock movie from the Cam-CAN dataset[26]. Five hundred seventy-seven adults (age range 18–88) where split into 34 groups, each consisting of participants within a specific age bracket (e.g., 18–23, 23–25, …, 83–88), to assess effects of age on neural state boundaries identified within each group. Data were hyperaligned[35] per group before applying GSBS to 5402 searchlights (average searchlight size of 97 voxels) covering the entire cortex. Neural state

boundaries were determined based on maximizing the correlation of timepoints within states and minimizing correlations of timepoints between states. This resulted in the optimal number and locations of neural state boundaries per searchlight for each age group. This optimal number of states is inversely related to the neural state duration as the duration of the movie stimulus was constant.

Our first aim was to see whether the state duration hierarchy that was observed previously in a younger sample (18–50 years[24]), is also present in this lifespan sample. We observed large variability in the median duration of neural states across brain regions (Fig. 1). In line with Geerligs et al.[24], we observed the shortest neural states in the visual cortex, caudate, and fusiform gyrus. The longest states were observed in higher-level regions such as the medial and lateral frontal gyrus, anterior cingulum, parahippocampal gyrus, and temporal pole, as well as in the cerebellum. Median state durations ranged from 4.9 s (2 TRs) in the searchlights with shortest states, up to 44.5 s (18 TRs) in the searchlights with the longest states. The cortical hierarchy pattern remains similar across the adult lifespan but with longer states across the cortex with age. For visualization purposes, we show this for the younger groups taken together, the middle groups, and the oldest groups (11, 12, and 11 subgroups, respectively).

### Increase in neural state duration with age

To quantify the effect of age on median state duration and variability in state duration we estimated the Spearman's rank correlation coefficient comparing the 34 age groups. There was a significant effect of age on median neural state duration in multiple regions across the cortex, with longer states with increasing age. The effect was strongest in the visual cortex and the ventromedial PFC (vmPFC; Fig. 2A). All 2816 significant searchlights (FDR-corrected) showed positive correlations between age and state duration. Direct comparison between the youngest and oldest groups confirmed this pattern, with 93.9% of significant regions showing longer durations in older adults (mean difference = 8.36 s, SD = 5.02, range −9.88–32.11; Supplementary Fig. 1). In the remaining 6.1% the difference between the youngest and oldest group was either neutral or negative but across groups all significant correlations were positive. Supplementary analyses (Supplementary Fig. 12) show that the association between age and state duration was mostly linear. We also investigated whether age was associated with the variability in state duration. To this end, we used a nonparametric version of the coefficient of variation, calculated by dividing the interquartile range (IQR) by median state duration to account for the typical increase of IQR with median duration. There was no effect of age on variability of state duration.

To visualize the effect of age on median state duration, we selected two searchlights in different clusters with the strongest effect of age, one in the

**Fig. 1 | Durations of neural states across the cortex averaged across the 11 youngest groups (top row), 12 middle-aged groups (middle row), and the 11 oldest groups (bottom row).** The optimal number of states varied greatly across regions, with shorter states in the visual cortex and longer states in the association cortex, such as the medial and lateral prefrontal gyrus. These patterns are similar for all three age groups with an overall increase of state duration across the cortex with age. The full range of median state durations across searchlights across 34 age groups was 2–18 TRs, however, averaging over groups resulted in a range of durations from 2.6 to 10.8 TRs. For visualization purposes, the color scale is limited to values between 4 and 9 TRs, as most searchlights fell within this range. Voxels with values outside this range are displayed using the color corresponding to the nearest bound (i.e., values < 4 TRs are shown as 4 TRs; values > 9 TRs are shown as 9 TRs).

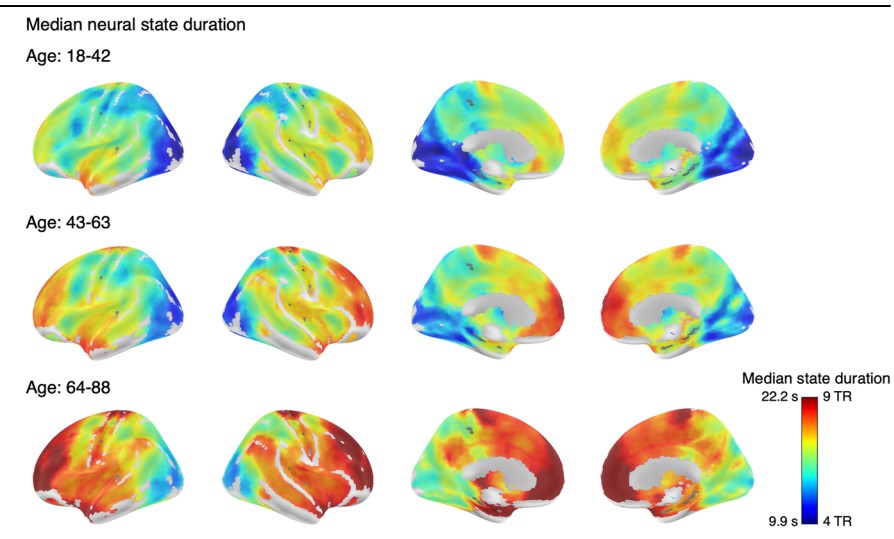

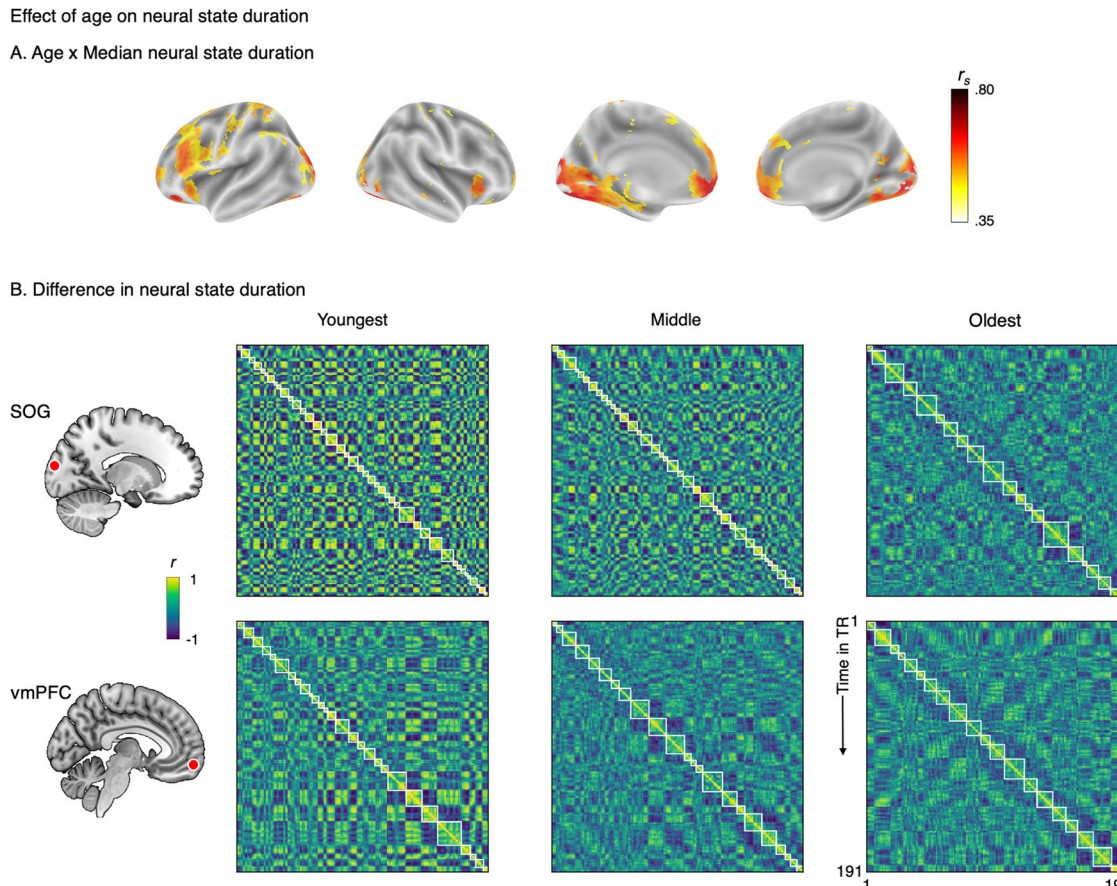

**Fig. 2 | Age affects neural state duration. A** The effect of age on median state duration thresholded at significance after FDR correction. State duration was longer with increasing age (based on 34 groups), especially in the visual cortex and vmPFC. Significant correlations at voxel level ranged from 0.38 to 0.80. **B** Locations of the searchlights that showed the highest correlation between age and median state duration on the left, followed by time-by-time correlation matrices (correlation based on the averaged brain activity time course in a searchlight for each TR) for the youngest, middle, and oldest group (left to right). White boxes represent the neural state boundaries detected by GSBS. The top row shows the data from the searchlight in the superior occipital gyrus (SOG, MNI −15, −96, 14), the bottom row from the vmPFC (MNI −7, 57, −13). It should be noted that longer neural state durations also necessarily imply that there are fewer distinct states, as the duration of the movie stimulus was constant. This is also apparent in the decreased number of neural states in the oldest group.

superior occipital gyrus (SOG; $r_s$ = 0.86) and one in the vmPFC ($r_s$ = 0.78), and plotted time-by-time correlations for the youngest, middle, and oldest groups (Fig. 2B, i.e., groups 1, 17, and 34 in Supplementary Table 1; see also Supplementary Fig. 1 for the relation between age group and number of states). Figure 2B shows an example of how with increased age neural states become longer. It also shows how correlations within states become weaker with age (i.e., less intense colors). With additional analyses and simulations, we show that this cannot fully explain the increased state duration we see in older adults (see sections on simulations and boundary strength).

### Widespread overlap between neural state boundaries and perceived event boundaries

Based on the principle of a nested cortical hierarchy[23,24], some neural state boundaries in earlier processing regions are expected to propagate to later levels of processing until they align with perceived event boundaries. We used the absolute overlap metric described by Geerligs et al.[24] to assess the overlap between neural state boundaries and perceived event boundaries. These perceived event boundaries were based on agreement of when one meaningful event ended and the next began in a separate group of 16 subjects[21]. The absolute overlap computes the total number of event and neural states boundaries that overlap and scales this with respect to the maximal and the expected number of overlapping boundaries (see Methods). Brain regions throughout the cortical hierarchy showed significant above chance overlap between neural state boundaries and perceived event boundaries (Fig. 3). The

overlap between neural state boundaries and event boundaries was the highest in the lateral and dorsal medial PFC, the superior frontal gyrus, frontal pole, Heschl's gyrus, and the insular gyrus. Additionally, we also found high overlap in the lateral visual cortex. The regions with the highest overlap are most likely to contribute to perceived event boundaries.

### Preserved overlap between neural states and events with age

We investigated whether the overlap between neural states and events differed across the lifespan. Spearman's rank correlation coefficient estimations showed that there was a minimal effect of age on the overlap between neural state boundaries and event boundaries. We found increased overlap in older adults in 4 voxels in the right middle temporal gyrus ($r_s$ = 0.65, Fig. 4A). These results suggest that aging does not substantially affect the association between neural state and event boundaries. In line with this, we found that the areas with the strongest effect of age on state duration are nonidentical to those with the strongest overlap between neural states and event boundaries (see Figs. 2 and 3). To quantify this, we computed the correlation across searchlights for the effects of age on state duration and the alignment between event and neural state boundaries. This correlation was −0.11 ($p < 0.001$), suggesting only a small overlap between the areas that show longer states with age and those that show an overlap with event boundaries. Additional network-level analyses showed distinct patterns of network involvement for these two processes (see Supplementary analysis 2, Supplementary Fig. 13).

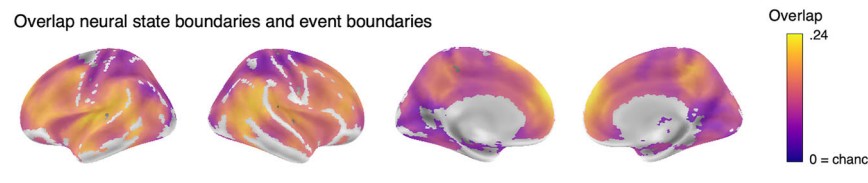

Overlap neural state boundaries and event boundaries

**Fig. 3 | Overlap between neural state and perceived event boundaries averaged across 34 groups.** The highest overlap at voxel level (0.24) was found in the superior frontal gyrus and insular cortex. The metric is scaled between zero (overlap based on chance) and one (all neural state boundaries overlap with an event boundary). Statistical significance is based on a ranksum test per searchlight (FDR corrected) to test if all the groups have an overlap different from 0.

Effect of age on overlap between neural state boundaries and event boundaries

A. Effect of age on overlap

B. Mean boundary presence separated by on event and off event TRs

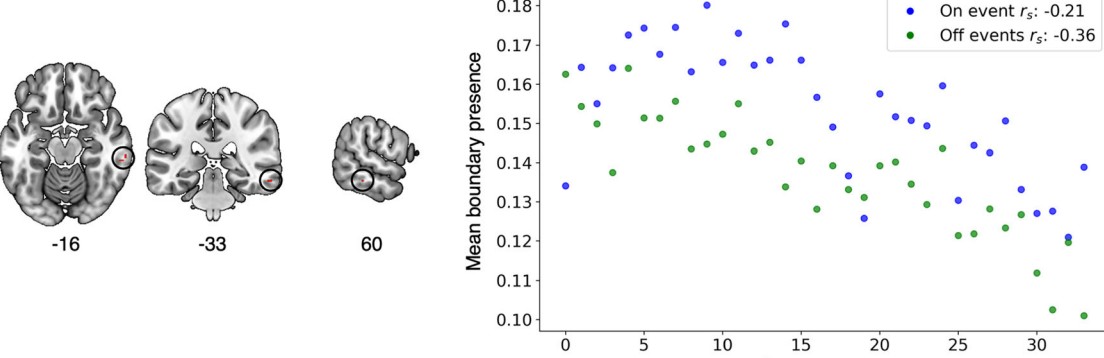

**Fig. 4 | The effect of age (34 groups) on the overlap between neural state and event boundaries. A** Four voxels in the right middle temporal gyrus show a significant *increase* with age on overlap ($r_s = 0.65$). FDR corrected for multiple testing.

Coordinates in MNI space. **B** Mean boundary presence across all searchlights per age group separated by on event boundary and off event boundary TRs.

To investigate if alignment with event boundaries made a difference for the effect of age on neural boundary occurrence, we examined the occurrence of neural states separately for TRs that overlap with event boundaries (on event boundary) and those that do not (off event boundary) across all searchlights. In both on event boundary and off event boundary TRs, we see an overall decrease in boundary occurrence with age (i.e., longer states), however this effect is stronger for off event boundary TRs (mean correlation across all searchlights $r_s = -0.36$) than on event TRs ($r_s = -0.21$; Fig. 4B, see also Supplementary Fig. 2). This explains why we observe longer neural states with increasing age without a decrease in overlap between event and neural state boundaries. Collectively, these findings suggests that the relationship between neural states and perceived event boundaries remains largely similar with age.

Even though averaging data across participants allows for greater signal-to-noise to identify neural states, it also has some drawbacks. One of them is that within-group inter-individual differences are obscured. It has been shown previously that estimating both the number and the location of state boundaries is especially challenging in single-subject data[28]. That is why we derived a fixed number of boundaries per participant based on the optimal numbers of states determined by GSBS of the whole group. To investigate the overlap between neural states and events on a single-subject level, we performed single-subject GSBS for two searchlights that showed the strongest overlap between events and neural states (both $r_s = 0.25$) and for two searchlights that showed the strongest effects of age on state duration (the same as those used in Fig. 2C for the effect of age, see Supplementary Fig. 3 for the location of all four searchlights). Next, we performed a Spearman rank correlation test between age as a continuous variable and the overlap between event boundaries and individual neural state boundaries.

The two searchlights in which overlap between event and neural state boundaries was high, showed a weak negative effect of age on overlap (superior temporal gyrus, STS: $r_s = -0.12$, $p < 0.01$; superior frontal gyrus,

SFG: $r_s = -0.12$, $p = <0.01$). In the searchlights with a strong effect of age on state duration, age did not significantly influence overlap between event and neural state boundaries (superior occipital gyrus, SOG: $r_s = -0.08$, $p = 0.06$; vmPFC: $r_s = -0.08$, $p = 0.06$). These single-subject results are in line with our results on the group level and confirm that the relationship between neural states and perceived event boundaries is largely preserved with age.

### Effects of age on state duration cannot be explained by noise or intersubject variability

Next, we wanted to investigate whether the effect of age on state duration could be due to increased variability among older participants or increases in noise. To this end, we performed a number of corrections on the empirical data, as well as different simulations.

The first of these was to control for inter-subject synchrony (ISS) in the association between state duration and age. ISS declines strongly with age[36,37], and is affected both by the inter-individual variability of participants, as well as by their signal-to-noise levels. ISS was computed for each subject compared to their age group and then averaged across the group. To test if the effects of age on median state duration could be attributed to variability or noise, we recomputed the correlation with age while controlling for ISS within each searchlight. We observed that 131 voxels still showed a significant increase in neural state durations with age (Fig. 5). These effects are primarily seen in clusters in the vmPFC and the visual cortex, suggesting that the age-related differences in neural state duration cannot be solely explained by increases in noise or increases in interindividual variability.

Additionally, we investigated the effect of age on ISS for each searchlight. Based on Spearman's rank correlation coefficient estimations, age had the strongest negative effect on ISS bilaterally in the temporal pole, the dorsolateral PFC, and the inferior parietal lobe (see Supplementary Fig. 4). This spatial pattern of correlations only had a small overlap with regions

**Fig. 5 | The effect of age (34 groups) on median state duration when including ISS as a covariate (after FDR correction).** Significant correlations between $r_s = 0.48$–$0.64$. Even with correction for ISS, median state duration was longer with increasing age in regions of the visual cortex and vmPFC. Coordinates in MNI space.

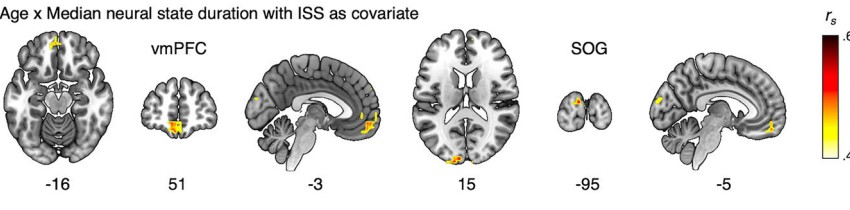

**Fig. 6 | The effect of age (continuous, 577 participants) on neural state boundary strength (after FDR correction).** Correlations between $-0.09$ and $-0.42$. Weakening of state boundaries is found across the brain, especially in the parietal and visual cortex.

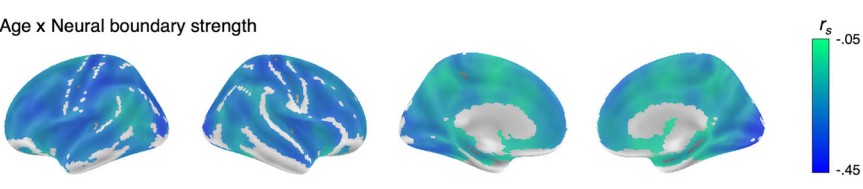

where the effect of age on neural state duration was strongest ($r_s = -0.09$, $p < 0.001$), indicating that age affects ISS and state duration largely in different regions of the brain. This suggests that there is a different driving force behind the age-related differences in neural state duration and the differences in ISS.

To look more specifically at whether the increased variability of neural state boundaries within each participant age-group was causing the observed increase in state duration, we also looked at neural state boundaries at the single-subject level. For the two searchlights that showed the strongest effect of age on state-duration, we performed single-subject GSBS. To get a pure measure of inter-subject variability in state boundaries, under the assumption that this might explain the increase in state duration, we fixed the number of neural states in each participant to the optimum across all participants. Within each age-group, we quantified the inter-subject variability in neural state boundaries by correlating the boundary timeseries of each subject, with the mean boundary timeseries of all other subjects in that age group. Next, we computed the average inter-subject boundary variability per subgroup and controlled for this in our correlation between age and state duration. As expected, we found that the overlap of boundaries across participants decreased strongly across the age groups (SOG: $r_s = -0.39$, $p = 0.02$; vmPFC: $r_s = -0.57$, $p < 0.001$). However, even after controlling for this decrease in overlap, we still observed a strong effect of age on state duration (SOG: $r_s = 0.83$, $p < 0.001$; vmPFC: $r_s = 0.71$, $p < 0.001$). Of course, these two searchlights showed a very strong and robust effect of age and state duration, potentially obscuring the confounding effect of inter-subject boundary variability. Therefore, we also investigated this for the searchlights that showed the strongest overlap between event boundaries and neural states. These searchlights also showed a decrease in overlap of neural state boundaries with age (STG: $r_s = -0.65$, $p < 0.001$; SFG: $r_s = -0.45$, $p = 0.008$). For these searchlights, only the SFG showed an effect of age on state duration (STG: $r_s = 0.15$, $p = 0.387$; SFG: $r_s = 0.50$, $p = 0.003$) and this association did not change after controlling for average inter-subject boundary variability (STG: $r_s = 0.20$, $p = 0.253$; SFG: $r_s = 0.46$, $p = 0.007$). Together, these results suggest that there is an increase in variability of neural state boundary locations with age. However, this variability cannot explain the increased duration of neural states that we observed with age.

To further investigate the confounds of variability and decreased signal-to-noise, we performed two sets of simulations. In one simulation, we investigated how interindividual temporal variability in the occurrence of state boundaries affected the duration of neural states by shifting neural state boundaries of the youngest group by one or two TRs. As can be seen in Supplementary Figs. 5 and 6, the time-by-time matrices look dissimilar from those of older adults. The number of states tend to decrease slightly with a simulated increase in inter-individual variability (see Supplementary Fig. 7). Importantly, these effects are not in the range that we observe with aging.

While our participant data shows that with advancing age, the number of states can decline very steeply (e.g. from 42 to 20 states in the SOG, and from 28 to 20 in the vmPFC, Supplementary Fig. 1), the simulations show a much smaller decrease (the largest difference is from 41 to 37 in the SFG), suggesting that increased variability in the timing of state boundaries cannot explain the observed increase in neural state durations with age.

In the next simulation, we investigated whether decreased signal-to-noise levels could explain the observed increase in neural state durations with age. To this end, we added increasing levels of random noise to the data of the youngest group to investigate if this could explain the increase in neural state durations with age. While visually this made the matrices look more similar to those of older adults (see Supplementary Figs. 8 and 9), we did not observe a systematic decrease in the number of neural states with higher levels of noise. Thus, increased noise with age is unlikely to be the cause of the observed increase in neural state durations.

Together, these simulations and empirical results demonstrate that increases in noise or inter-individual variability cannot explain the observed increase in state durations with age.

## Weaker neural state boundaries only partly explain the effect of age on state duration

Theoretically, neural state boundary strengths can vary between just above 0 and 2 (1 minus the correlation between the neural activity patterns of two adjacent states), with higher values indicating that two successive states are more dissimilar. We investigated whether the observed increase in neural states was accompanied by a weakening of state boundaries with age. Weaker boundaries may be more difficult to detect with GSBS, which could also result in the appearance of longer neural states. To test this, we first identified the neural state boundaries in data averaged across all participants across the lifespan. For this common set of boundaries, we computed the neural state boundary strength in each individual participant. We found that Spearman's rank correlation coefficient estimations, boundary strength was strongly negatively correlated with age (i.e., the pattern of neural activity changes less at state boundaries with increasing age, age is here at the individual level; see Fig. 6). This effect was strongest in visual areas including the occipital pole and the fusiform gyrus. Other areas with a weakening of state boundaries were the supramarginal and postcentral gyrus/somatosensory cortex, insular cortex, and middle temporal gyrus. This shows that indeed neural state boundaries were weaker in older adults and that this might explain part of the observed lengthening of neural states with age. There was only limited overlap between regions where age affects state duration and boundary strength most ($r_s = -0.16$, $p < 0.001$). In addition, we observed that the effect of age on state duration persisted in the vmPFC, even if we used the average state boundary strength within each subgroup as a covariate (see Supplementary Fig. 10). Additionally, we tested for different relationships between age and state duration and strength (see

Supplementary analysis 1, Supplementary Fig. 12) and found that whereas state duration linearly increases with age, boundary strength shows a quadratic relationship on top of a general decline. Taken together, these findings suggest that the lengthening of neural states could not be fully explained by the weakening of neural states with age.

Weaker boundaries might result from decreased similarity within neural states (resulting in a less coherent neural pattern per state or more noisy data) as well as from decreased distinction of neural patterns across states. Therefore, we also tested whether the correlation between age and boundary strength remained after correcting for average within state correlation (as reduced within-state coherence could give the impression that boundary strength decreases with age). However, the effect of age on boundary strength remained, suggesting that weaker state boundaries with age cannot be explain by a lack of coherence within states (see Supplementary Fig. 11). Although it should be noted that within-state correlations also declined with age (see Supplementary Fig. 11).

## Discussion

Perceived events are represented in the brain as a series of distinct neural states. The present study aimed to determine whether these neural states, and their correspondence with perceived events, differ with age. Using movie-based data from a large cohort spanning the adult lifespan, combined with a data-driven state segmentation method, we demonstrate that older adults have longer neural states, while maintaining a similar cortical organization of state duration. In the visual cortex, where the shortest states were observed, we found a strong effect of age on state duration. Increased state duration with age was also prominent in the vmPFC, an area with the longest states. Importantly, the areas with the highest overlap between neural states and perceived event boundaries are dissimilar from those with the strongest effect of age on state duration and the overlap between neural states and events was preserved with age. This suggests that coarse event segmentation largely remains stable across the lifespan. Simultaneously, our results suggest that aging is associated with neural temporal dedifferentiation, or a blurring of time, and changes in visual and affective processing.

### Neural state duration

Previous findings have suggested that neural states may be organized in a nested cortical hierarchy[23]. This has recently been demonstrated across the entire cortex in a sample of adults aged 18–50[24]. Our results extend that finding to the full adult lifespan. We show a similar pattern of temporal cortical hierarchy: short neural states in the visual cortex, caudate, and fusiform gyrus, while the longest states were observed in higher-level regions such as the medial and lateral frontal gyrus, anterior cingulum, parahippocampal gyrus, and temporal pole. Short states might be directly related to perceptual features[38]. Areas like the medial prefrontal cortex and middle frontal gyrus have been associated with different higher level cognitive processes[39–41]. In the context of movies, they may be related to the narrative and to memory processes, attention, and goals of the viewer[42]. Neural states in these regions last up to 30 s which allows for integration of information over a longer time span. We find that age increases neural state duration both in regions with the shortest states, the visual cortex, and the longest states, the vmPFC.

### Longer and less distinct neural states with age

Our results show a strong effect of age on the duration of neural states, especially in the visual cortex and vmPFC. Neural state duration in the visual cortex has been related to perceptual processing of visual changes in a movie[38] which changes with age[43]. One of the processes the vmPFC is involved in is future simulation and integration of schematic knowledge[40,44]. The medial PFC is also a key node in the default mode network, which in aging has been related to reduced suppression of representations and increased use of associations and knowledge of the world[45].

We discuss potential mechanisms for the observed increase in neural state duration with age. First, this increase in state length might seem reminiscent of age-related slowing[46]. According to this view, all cognitive and neural processes are slowed with age, which leads to incremental delays further up the information processing hierarchy. However, a delay in processing time would not necessarily lead to longer states, especially for regions with longer state durations, where age-related slowing in the range of hundreds of milliseconds[47–49] could not affect state durations in the range of 20 s.

Another possible mechanism is reduced inhibitory control with age[50]. Previous work suggests that older adults maintain access to previously attended information that is no longer relevant to the task at hand and that this is related to perceptual processing areas[31–33] (review[34]). If older adults still have the previous event in mind as the next event starts, this could lead to some missed state boundaries and/or weaker boundary strengths with age. Indeed, we observed a decrease in number of boundaries and boundary strength with age in different regions of the visual cortex; that is, older adults showed a longer state duration and less change in the pattern of neural activity between successive states (Figs. 2 and 6). While the precise cause of this temporal dedifferentiation remains unclear, it is possible that reduced inhibition leads to the blurring of events over time. This fits with recent behavioural evidence suggesting that older adults with poorer inhibitory control show relatively stronger associations *across* event boundaries when recalling information from long-term memory[51].

The observed temporal dedifferentiation across events could also be related to the spatial dedifferentiation of neural representations that has been observed across the lifespan. With advancing age, neural representations (i.e. patterns of voxel activity) of different items and different stimulus categories become less distinct[29,30]. Since different moments in time in the movie effectively show different collections of stimuli, dedifferentiation of neural representations of different stimuli with advancing age would translate into increased similarity in neural representations over time. Therefore, the temporal dedifferentiation we see as expressed in weaker boundaries between states or fewer distinct states in the visual cortex could be a direct consequence of spatial dedifferentiation.

There are several potential explanations for spatial dedifferentiation, which may by extension also explain the temporal dedifferentiation we observe. Here we will mention two of these explanations which have received some empirical support over the last couple of years. The first is that declines in the γ-aminobutyric acid (GABA) inhibitory neurotransmission may underlie dedifferentiation[52,53]. This has been shown to play a role in single-neuron selectivity in animal work. Recent evidence in humans has shown that GABA levels relate to neural distinctiveness of auditory and visual stimuli in younger and older adults. The second account is the lifetime experience hypothesis[29,30], which suggests that the increase in perceptual experience and knowledge over the lifespan increases older adults' ability to assimilate new exemplars into pre-existing perceptual schema. Consequently, with increasing age, processing of novel exemplars will show a more similar neural representation to processing of previously experienced exemplars. Taken together, decreasing neural inhibition and increased knowledge and experience could be an explanation for both spatial and temporal dedifferentiation in aging.

An interesting question is how longer and less distinct neural states with age may affect perception and memory in everyday life. Previous research has shown that perception of time passing is related to an accumulation of salient physical events in a movie[54], as well as to an accumulation of salient events in BOLD responses[55,56]. This suggests that longer (fewer) neural states within the same period may contribute to older adults experiencing time as passing more quickly, which is in line with findings from previous studies (John & Lang, 2015). Furthermore, Baldassano et al.[23] have associated neural state boundaries to storing of the current event model in memory, and we recently showed that greater distinction between successive neural states relates to better episodic memory in both younger and older adults[57]. Therefore, longer and less distinct neural states with age might influence the formation of memories for events in the movie. This hypothesis needs to be tested in future research.

## Preserved coarse event segmentation

Neural state boundaries throughout the entire hierarchy overlapped with perceived event boundaries. This overlap is particularly strong in the lateral and dorsal medial PFC, the superior frontal gyrus, frontal pole, Heschl's gyrus, and the insular gyrus. This finding expands on the previous study[24] showing a similar pattern of overlap in a subsample of subjects aged 18–50. Areas with high overlap most likely contribute to the conscious experience of event boundaries. Perceived event boundaries have previously been shown to align with changes that are relevant to the story line, for example changes in characters, spatial locations, and goals of actors[9] which might explain why overlap is seen across the temporal hierarchy.

Results regarding the effect of age on event segmentation are somewhat mixed. It has been suggested that there might be a difference between younger and older adults in fine (short time-scale) but not in coarse (longer time-scale) event segmentation, with older adults reporting longer events in the fine event segmentation condition[10,58]. In contrast, Sargent et al.[6] did not find age differences in either fine or coarse event segmentation. Using the same movie as in the current study, Reagh et al.[18] found no differences in segmentation between younger and older adults' coarse event boundaries. This lack of an age difference is in line with our finding that age has minimal to no effect on the correspondence between neural state and event boundaries. It is important to note here that in quantifying this correspondence, we looked at the proportion of neural state boundaries that overlap with an event boundary. This means that even if participants differ in their total number of neural state boundaries, they can still show similar proportions of overlap. Our findings therefore suggest that, among the neural state boundaries that do occur, their tendency to align with event boundaries is similar across younger and older participants. However, this does not imply that the actual number of event boundaries that align with neural state boundaries is the same across age groups. Importantly, the areas with the strongest correspondence have minimal overlap with those regions showing the greatest increase in neural state duration with age. Network-level analyses revealed distinct patterns for the two processes examined (see Supplementary Analysis 2, Supplementary Fig. 13). While age-related increases in neural state duration were most pronounced in the visual network, the correspondence between perceived event boundaries and neural state boundaries was highest in attention-related networks (primarily salience/ventral attention). This network-level distinction suggests that aging affects neural state dynamics primarily in visual processing regions (fitting with the above-described findings on inhibitory control), while event boundary detection might rely on networks involved in salience processing and detection of relevant information from the environment[59].

The strongest overlap between neural state and event boundaries was observed in the dmPFC, which in previous studies has been linked to knowledge about scripts of typical event sequences[60]. This is in line with behavioural work showing that knowledge and event schemas are important determinants of event perception[61–63]. Since crystallized intelligence (i.e., knowledge) tends to be either preserved or increase throughout the lifespan[64–66] and knowledge of event scripts is preserved[6], prior knowledge may be an important reason why there does not appear to be a substantive decline in coarse event segmentation across the adult lifespan.

## Validation of our results

Our results show that increasing age is associated with longer neural states in the visual cortex and vmPFC. We demonstrate that this finding cannot simply be explained higher variability or increased noise with age. In line with previous studies, we saw a decrease in ISS with age[36], but whilst controlling for ISS we still saw increases in neural state duration predominantly in clusters of the visual cortex and vmPFC. Furthermore, artificially adding noise or temporally shifting the data of the youngest subjects did not result in patterns similar to those of older adults. Finally, we investigated if the increase of state durations could have been caused by the disappearing of neural boundaries that are weak in the first place. We found that with age neural state boundaries were weaker which might explain part of the

observed lengthening of neural states. However, we found that it could not completely account for the observed association between age and state duration. Thus, increased neural state duration and weakening of neural boundary strength with increasing age seem at least partly differentiable. Previous studies have documented age-related changes in the hemodynamic response function; however, it is unlikely that the age-related increase of several seconds in neural state durations observed in our study can be explained by these changes, as they are typically less than one second in magnitude[67]. For a more extensive discussion on limited effects of hemodynamic response delays on neural state duration estimates, see Geerligs et al.[24].

## Limitations and future directions

Using naturalistic stimuli, like movies, compared to highly controlled typical lab-based tasks means that there are some task confounds that are not controlled for (e.g., our use of a single film). At the same time, it offers new possibilities. This study and previous work[23,24] demonstrates how movie data can be used to investigate the processing of continuous stimuli without an explicit task, akin to daily life. Especially when comparing groups of different ages or studying patient populations, naturalistic stimuli have the advantage of reducing group differences caused by differential understanding or experience of the task demands[68].

A potential limitation is that subjective event boundaries were defined using data from an independent sample of 16 younger adults[21]. However, Reagh et al.[18] found no significant differences in event segmentation agreement between 14 older and 14 younger adults for the same movie. Additionally, recent work has demonstrated that stable event boundary agreement can be achieved with relatively small samples[69]. Finally, regions showing the strongest age-related effects in our data had limited overlap with those showing the greatest alignment between neural states and subjective boundaries. Together, these findings suggest that age-related differences in event boundary perception are unlikely to account for our results.

We found that duration of neural states increases with age in visual cortex and vmPFC. This seems related to neural boundaries that disappear with age, especially those that do not align with perceived event boundaries. Future studies should determine if the boundaries that disappear with age are those that align with specific types of events. These could be event boundaries that have lower interrater agreement across participants on an event segmentation task or changes in specific aspects of the stimulus like spatial location or goals of the actor.

More broadly, much is still unknown about what neural states represent and how individual differences in neural states might shape differences in event segmentation and related cognitive processes like perception, action and memory. This makes it difficult to make strong claims about how the age-differences we observed here affect the processing of events. More generally though, our approach offers a way to investigate the changes in segmentation on the neural level, which are a strong neural correlate of event segmentation, without altering the mental state of the participant by instructing them to segment the movie into meaningful units. As such, it offers a steppingstone for follow-up research to identify exactly how the temporal dedifferentiation of neural states affects how events are segmented, represented and memorized.

## Conclusion

Our results show a cortical hierarchy of neural states during movie viewing in a large lifespan cohort. We demonstrate that neural state durations were longer in older than younger adults in low-level visual processing regions and in high-level prefrontal regions related to suppression of information and integrating knowledge. This results in decreased temporal differentiation between experiences in close temporal proximity. This may be a sign of efficiently mapping new information onto known event schemas, but it may also result in blurring of details across events. Critically, the relationship between neural states and perceived event boundaries remained similar with age. This suggests that neural state boundaries that may underlie the experience of major (coarse) events remain relatively stable across the adult

lifespan. This is in line with behavioural findings that show that older adults identify the same coarse event boundaries as younger adults.

## Methods

### Participants
This study reports on data from 577 adults (293 females) who were aged 18–88 (mean age 53.39, SD = 18.42) from the healthy, population-based cohort tested in stage II of the Cam-CAN project[26,27]. This subset of participants was also used in a previous study on age-related differences in intersubject synchronization of brain activity[37]. Participants had English as a first language, normal or corrected-to-normal vision and hearing, no contraindications to MRI, and no neurological disorders[26]. All ethical regulations relevant to human research participants were followed. Ethical approval for the study was obtained from the Cambridgeshire 2 (now East of England – Cambridge Central) Research Ethics Committee. Participants gave written informed consent.

### Movie
Participants watched a shortened version of a black-and-white television drama by Alfred Hitchcock called "Bang! You're Dead" while undergoing an fMRI scan. The full 25-minute episode was shortened to 8 min with the narrative preserved[26]. Previous studies have shown that this shortened version of the movie elicits robust brain activity, synchronized across participants[36,37]. Participants were instructed to watch, listen, and pay attention to the movie.

### Event boundaries
Perceived event boundaries for "Bang! You're Dead" were identified by a separate group of subjects[21]. Sixteen observers watched the same shortened movie outside the scanner and indicated with a keypress when they felt 'one event (meaningful unit) ended and another began.' Boundaries of different subjects were taken together if they were close in time, and those identified by a minimum of five participants were marked as "true" boundaries, which resulted in 19 boundaries, separated by 6.5–93.7 s. Across the 8-minute-long video, these boundaries tended to coincide with changes in coarse-grained events (i.e., large temporal and spatial changes).

### Image acquisition and preprocessing
MRI scanning (details as in e.g., Geerligs et al.[37]) was done on a 3T Siemens TIM Trio System at the MRC Cognition Brain and Sciences Unit, Cambridge, UK. For the fMRI sequence, 193 volumes of movie data were acquired with a 32-channel head-coil, using a multi-echo, T2 ∗ -weighted echo-planar imaging (EPI) sequence. Each volume contained 32 axial slices (acquired in descending order), with slice thickness of 3.7 mm and interslice gap of 20% (repetition time (TR) = 2470 ms; five echoes [TE = 9.4 ms, 21.2 ms, 33 ms, 45 ms, 57 ms]; flip angle = 78°; field-of- view = 192 × 192 mm; voxel-size = 3 × 3 × 4.44 mm), the acquisition time was 8 min and 13 s. A high-resolution (1 mm isotropic) T1 - weighted image was additionally acquired.

The details of initial steps of data preprocessing for the movie data are also described in Geerligs and Campbell, 2018. Analysis of functional neuroimages software (AFNI; version AFNI_17.1.01; https://afni.nimh.nih.gov) and the statistical parametric mapping software (SPM12; (http://www.fil.ion.ucl.ac.uk/spm) were used for preprocessing. The AFNI parts of preprocessing included deobliquing of each TE, slice time correction, realignment of each TE to the first TE in the run, and multi-echo independent component analysis (ME-ICA) denoising. ME-ICA is a method for removal of non-BOLD-like components from the fMRI data, including effects of head motion[70]. In SPM, the ME-ICA denoised data were co-registered, followed by DARTEL intersubject alignment which allows for transformation to an age-representative template that is subsequently transformed to Montreal Neurological Institute (MNI) space. For 30 participants, there was either a problem with running the ME-ICA denoising or a problem with the normalization; these participants were not included in further analyses. Furthermore, participants with a high level of mean head motion (2 SDs

above the mean, 32 participants) and participants for whom nearly all components were removed during the ME-ICA denoising (>88% of all components, 2 participants) were not included in the analyses. This resulted in a final sample of 577 participants described under "participants". The data were highpass-filtered with a cut-off of 0.008 Hz before further analyses. We did not apply hemodynamic response function deconvolution to the data before detecting neural states. This is because deconvolution was found to be beneficial for detecting very brief states of 1 or 2 TRs, but unfortunately also reduces the distinctiveness of neural state boundaries due to lower signal-to-noise ratio in the data[24].

### Hyperalignment
Prior to neural state segmentation, to optimally align voxels across participants of each group, we used whole-brain searchlight hyperalignment as implemented in the PyMVPA toolbox[35,71]. Hyperalignment is an important step in the pipeline because the neural state segmentation method relies on group-averaged voxel-level data. Hyperalignment uses Procrustes transformations to derive the optimal rotation parameters that minimize intersubject distances between responses to the same timepoints in the movie. The details of the procedure are described in Geerligs et al.[28]. Here, we applied hyperalignment to groups of participants, such that the same groups of participants that were averaged prior to neural state detection were first hyperaligned together (see next section for a description of these groups).

### Neural state segmentation (GSBS)
Single-participant data is too noisy to reliably identify both the number and the locations of neural state boundaries[28]. Therefore, we averaged the data within 34 subgroups of different ages (each consisting of 17 participants of a similar age) to investigate the effects of age on neural state duration. This provided sufficient signal-to-noise to be able to estimate both the number and location of neural state boundaries, while also leaving a sufficient number of groups to estimate age effects (group size based on Geerligs et al.[24]; see Supplementary Table 1 for participant demographics per group). This is comparable to the approach in Cohen et al.[72], where developmental differences in neural state boundaries were also investigated in group-averaged data.

Within each age-group, the fMRI data were hyperaligned and subsequently averaged within each group before GSBS was used to estimate neural state boundaries. The steps to identify neural state boundaries in a data driven way are identical to Geerligs et al.[24]. In short, we used Greedy State Boundary Search (GSBS[28]) which performs an iterative search for state boundary locations that optimize the similarity in average activity patterns within a neural state and maximal dissimilarity in consecutive states. The input to the GSBS algorithm consists of a set of voxel time courses within a searchlight. For the main analyses, spherical searchlights were scanned within a group specific mask with a step size of two voxels and a radius of three voxels. To define the mask, a threshold was set at 70% of the mean activity across all voxels for each participant. Voxels meeting this threshold in all participants were included. The final mask only included voxels overlapping with the grey matter SPM template, thresholded at >0.2 probability. This resulted in 5204 searchlights. Searchlights had an average size of 97 voxels (max: 123; IQR: 82–115); variation in size was a consequence of exclusion of out-of-brain voxels. Searchlights had to have a minimum size of 15 voxels to be included in the analysis. Data from searchlights were projected to voxels for plotting on the brain. We used the improved version of GSBS with better reliability and validity as described in Geerligs et al.[24].

For an additional control analysis described in the sections below, we also performed state detection after averaging the data from all participants together, without dividing the data into separate subgroups. In this case, hyperalignment was applied to all participants together.

### Metrics of interest and statistical tests
The first step in our analysis was to investigate whether this life-span sample showed similar patterns of neural state durations as our previous work in a

younger sample[24]. To do this, we first computed the metric of interest (i.e., median state duration and variability in state duration) per age group before averaging the data across age-groups to visualize patterns in younger, middle, and older adults (11, 12, and 11 groups, respectively). As the variability in state duration metric tends to increase with median duration, we used a nonparametric version of the coefficient of variation, calculated by dividing the interquartile range (IQR) by median state duration.

Our main aim was to examine the effect of age on neural states. Age effects on median state duration and variability in state duration were estimated with the Spearman's rank correlation coefficient comparing the 34 age groups. Analyses were FDR corrected across searchlights. For visualization of the effect of age on median state duration, searchlights with the strongest effect of age were selected and time-by-time correlations matrices were plotted for the youngest, middle, and oldest groups.

To assess the overlap in neural state boundaries and perceived event boundaries, we used the absolute overlap metric described by Geerligs et al.[24]. The absolute overlap computes the total number of event and neural states boundaries that overlap and subsequently scales this with respect to the total number of boundaries as the maximum value and the expected number of overlapping boundaries as the minimum value. A measure of one indicates that all neural state boundaries (per searchlight, per age group) align with an event boundary (same for all searchlights and age groups), zero indicates an overlap as expected by chance. For example, imagine there are 20 neural states boundaries, 10 event boundaries and 100 time points, of which 8 overlap. In this case the expected overlap would be 20/100*10/100*100 = 2. The absolute overlap would be (8-2)/(20-2) = 0.334. In practice, the scaling with respect to the expected overlap provides an appropriate baseline, but it does not affect the observed differences between regions or groups. This means that all regional and age-related variability reported in the paper is identical compared to simply looking at the percentage of all neural state boundaries that overlap with an event boundary. Therefore, if one group has 20 boundaries, of which 10 overlap with event boundaries and the other group has 30 boundaries, of which 15 overlap with event boundaries, these groups will have a nearly identical absolute overlap.

To account for the hemodynamic response in the BOLD signal, we added 5 s to the onset of each perceived event boundary, and then allowed for a 1-s window around that timing to account for slight variability in alignment with TRs (some perceived event boundaries are at the boundary of two TRs), finally we converted seconds to TRs (i.e., divided by 2.47). To calculate the significance of the overlap, we tested with a ranksum test whether the boundary overlap was significantly different from zero across the 34 age groups after false discovery rate (FDR) correction. To visualize patterns, we averaged that overlap in younger, middle, and older adults (11, 12, and 11 groups, respectively). The effect of age on absolute overlap was estimated with the Spearman's rank correlation coefficient comparing the 34 age groups, FDR corrected across searchlights. Subsequently, using a correlation test, we assessed the spatial overlap between the effects of age on state duration and on neural state boundary correspondence to perceived event boundaries to determine if these effects are observed in the same brain regions.

Finally, we examined if age affected neural state boundary occurrence differently for TRs that overlap with a perceived event boundary versus those that do not. For each TR, boundary occurrence across age groups was calculated. TRs were split into those that overlapped with a perceived event (allowing a 1-s window), and those that were non-event TRs. The effect of age on neural boundary occurrence was estimated with Spearman's rank correlation coefficients in each of these types of TRs. This was done for the average across all searchlights, as well as for each searchlight separately.

### Additional analyses and simulations

Even though averaging data across participants allows for greater signal-to-noise to identify neural states, it also has some drawbacks. For example, by averaging data across participants, increased individual variability in the timing of boundaries may result in the disappearance of these boundaries, and as a result, an apparent increase in state duration after averaging across

these individuals. This is similar to how averaging across trials may obscure interesting variability across trials, or even result in incorrect inferences of brain activity (Stokes and Spaak, 2016). Higher noise levels in older compared to younger age-groups, for example due to cardiovascular factors[73] could also impact the group-level estimates.

To address these potential confounds, we performed a number of additional analyses. First, we derived single-subject neural state boundaries with a fixed number of boundaries per participants. We did this for four pre-selected searchlights (2 with strong overlap between event and neural state boundaries, and 2 with a strong effect of age on state duration). This allowed us to investigate: 1) Whether the effect of age on neural state duration could be explained by increased variability in neural state boundary locations across the lifespan and 2) Whether these single subject results provide support for the lack of age-effect in the overlap between events and neural states. Because single-subject fMRI data suffers from low signal to noise ratio, which can affect the inference of the number of boundaries with GSBS, we pre-set the number of boundaries in each participant to be equal to the optimum that was derived based on the whole group. To answer question 1, we computed the average Pearson correlation between the boundary timeseries in each participant and the averaged timeseries of all other participants in the same subgroup. This resulted in one average correlation per subgroup, which we then used to test whether increased variability in boundary location with age can explain the age-related increase in state durations. To answer question 2, we estimated the Spearman's rank correlation coefficient between age (as continuous variable) and the overlap between event and neural state boundaries.

In the second analysis, we looked at interindividual variability in a different way. We used inter-subject synchrony (ISS) as a confound regressor. ISS measures the alignment between each participant's brain data and that of the rest of the group, and therefore decreases when there are differences in timing across participants (including differences in neural state boundaries) and when there are increased levels of noise. Unlike low-level signal quality metrics such as temporal signal-to-noise ratio (tSNR), ISS captures both noise-related reductions in signal stability as well as inter-individual differences in time-locked neural processing. In previous work, we have shown that ISS declines strongly with age[36,37]. Thus, we thought it important to demonstrate that age differences in neural states persist after controlling for ISS. To this end, we computed ISS by calculating for each subject the correlation of the time courses of that subject with the mean of all other subjects in the same age group and subsequently taking the mean ISS of all subjects in that age group to get an average ISS value per group. To test whether this ISS decline could explain the effect of age on median state duration, we performed a partial correlation between age group and median state duration, controlling for ISS, using the Spearman's rank correlation test. Additionally, we assessed the effect of age on ISS for each searchlight.

Third, we performed simulations, in which we manipulated either the amount of noise in the data or the amount of inter-individual variability in the state boundaries. We applied these simulations to the data from the youngest group of participants for the same two pre-selected searchlights used in Fig. 2 (with the strongest effect of age on state duration) and two with the strongest overlap to see if this emulated the effect of age. In the first set of simulations, we added random noise to the data of each voxel with three different levels of variability (5, 50, and 100% of the standard deviation in the original signal). We then reran GSBS and computed the number of states and visualized time-by-time correlation matrices to compare with those of the actual data of the older groups. The simulation was repeated 100 times to obtain a range of estimates of how noise affects the number of neural states. In the second set of simulations, we increased the intersubject variability in the timing of neural state boundaries. We simulated a situation where one half of the subjects had a neural state boundary one or two TRs before the other half of the subjects. To this end, we took the group-level data of the youngest group, shifted it by one or 2 TRs and averaged it together with the unshifted (original) data (after removing the 2 first and last TRs from the data to keep the same length). We reran GSBS for to determine the optimal number of states for each of the shifted datasets and we compared the

stimulated number of states and time-by-time correlation matrices with the older group data.

Finally, a potential explanation for our findings is that boundaries weaken with age, and that due to this weakening of boundaries, some boundaries are no longer detected by GSBS, which results in the appearance of longer neural states. To address this question, we performed an additional analysis in which we investigated differences in strength of neural state boundaries at the single subject level. To compute individual boundary strength using GSBS, the first step was to determine a shared set of boundaries by applying neural state segmentation to the entire group of participants. Next, we investigated the strength of these boundaries within each individual. Here, boundary strength is defined as 1 minus the Pearson correlation of the average neural activity patterns of the two neural states surrounding each boundary. Thus, a higher strength implies more dissimilarity between two consecutive neural states. The effect of age on neural boundary strength was estimated with Spearman's rank correlation coefficients, as strength was not normally distributed. A second correlation test was used to assess the overlap in brain regions where age affected boundary strength and state duration most. Weaker boundaries might result from decreased similarity within neural states (resulting in a less coherent neural pattern per state or more noisy data) as well as from decreased distinction of neural patterns across states. Therefore, we also investigated if the correlation with boundary strength remained even if we corrected for the average within state correlation, as weaker time-by-time correlations overall could make it seem like boundaries becomes less distinct with age.

## Statistics and reproducibility

Participants were divided into 34 age groups for analysis, unless otherwise specified. Neural states were identified using the data-driven GSBS method, and age effects were assessed with Spearman's rank correlation coefficients, using age group as the independent variable. Where age was treated as a continuous variable at the individual level, this is explicitly stated. Full details of the statistical analyses are provided in the preceding sections. The code used to generate the results in this paper is available at https://osf.io/4z67t/ and https://github.com/slugtmeijer and additional code at https://github.com/lgeerligs. The GSBS algorithm used is released in a Python package: https://pypi.org/project/statesegmentation/. Surface plots are illustrative and volumetric .nii files generated by the provided code are shared on OSF and GitHub.

## Reporting summary

Further information on research design is available in the Nature Portfolio Reporting Summary linked to this article.

## Data availability

The data used in this project can be requested via - https://camcan-archive.mrc-cbu.cam.ac.uk/dataaccess/.

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

## Acknowledgements

The Cambridge center for Ageing and Neuroscience (Cam-CAN) was supported by the UK Biotechnology and Biological Sciences Research Council (grant number BB/H008217/1), together with support from the UK Medical Research Council Cognition & Brain Sciences Unit (CBU) and University of Cambridge, UK. The authors are grateful to the CamCAN respondents and their primary care teams in Cambridge for their participation in this study. The authors also thank colleagues at the MRC Cognition and Brain Sciences Unit MEG and MRI facilities for their assistance. The Cam-CAN corporate author consists of the project principal personnel: Lorraine K Tyler, Carol Brayne, Edward T Bullmore, Andrew C Calder, Rhodri Cusack, Tim Dalgleish, John Duncan, Richard N Henson, Fiona E Matthews, William D Marslen-Wilson, James B Rowe, Meredith A Shafto; Research Associates: Karen Campbell, Teresa Cheung, Simon Davis, Linda Geerligs, Rogier Kievit, Anna McCarrey, Abdur Mustafa, Darren Price, David Samu, Jason R Taylor, Matthias Treder, Kamen A Tsvetanov, Janna van Belle, Nitin Williams, Daniel Mitchell, Simon Fisher, Else Eising, Ethan Knights; Research Assistants: Lauren Bates, Tina Emery, Sharon Erzinçlioglu, Andrew Gadie, Sofia Gerbase, Stanimira Georgieva, Claire Hanley, Beth Parkin, David Troy; Affiliated Personnel: Tibor Auer, Marta Correia, Lu Gao, Emma Green, Rafael Henriques; Research Interviewers: Jodie Allen, Gillian Amery, Liana Amunts, Anne Barcroft, Amanda Castle, Cheryl Dias, Jonathan Dowrick, Melissa Fair, Hayley Fisher, Anna Goulding, Adarsh Grewal, Geoff Hale, Andrew Hilton, Frances Johnson, Patricia Johnston, Thea Kavanagh-Williamson, Magdalena Kwasniewska, Alison McMinn, Kim Norman, Jessica Penrose, Fiona Roby, Diane Rowland, John Sargeant, Maggie Squire, Beth Stevens, Aldabra Stoddart, Cheryl Stone, Tracy Thompson, Ozlem Yazlik; and administrative staff: Dan Barnes, Marie Dixon, Jaya Hillman, Joanne Mitchell, Laura Villis. Linda Geerligs was supported by a VIDI grant of the Netherlands Organization for Scientific Research (grant number VI.Vidi.201.150). Karen L. Campbell was supported by the Natural Sciences and Engineering Research Council of Canada (Grant RGPIN-2024-03800), the Canadian Institutes of Health Research (Grant PJT 180591), and the Canada Research Chairs program.

## Author contributions

CRediT S. L.: Conceptualization, Formal analysis, Methodology, Software, Visualization, Writing – original draft. D. O.: Methodology, Software, Visualization, Writing – review and editing. L. G.: Conceptualization, Data curation, Formal analysis, Methodology, Software, Supervision, Validation, Visualization, Writing – review & editing. K. L. C.: Conceptualization, Data curation, Funding acquisition, Project administration, Resources, Supervision, Writing – review & editing.

## Competing interests

The authors declare no competing interests.
