## [Transparent Peer Review file · Communications Biology]

Temporal dedifferentiation of neural states with age during naturalistic viewing

Corresponding Author: Dr Selma Lugtmeijer

Version 0:

Reviewer comments:

Reviewer #1

(Remarks to the Author)

This manuscript describes an extension of a secondary analysis of an existing dataset previously reported by the same group. Here, whole brain fMRI data that had been continuously acquired during the viewing of a short movie was analyzed to segment the fMRI timeseries into distinct 'states'. The key topics of interests were regional differences in the duration of different brain states, whether and where state transitions overlapped with a separate sample's judgments of event-transitions in the movie itself, and the effects of age on these variables. It was reported that state duration was positively correlated with age in regions demonstrating both short (visual cortex) and long (PFC) state durations, but that there was no effect of age on the extent that state transitions mapped onto reported event boundaries in the movie (these transitions were confined to regions where state duration was little affected by age). It is concluded that the age differences in state duration that were identified are a reflection of age-related 'temporal dedifferentiation', and that the event segmentation results are consistent with reports of null findings for the effects of age on event boundary judgments.

The results extend the authors' original report and will be of interest to those interested in age differences in brain dynamics and in perceptual and event representations. The paper would benefit from attention the following issues:

- i) Some of the conclusions about age differences appear to be based on visual inspection rather than statistical analysis. For example, the assertion that the correlation matrices illustrated in figure 3 differ according to age does not seem to be supported statistically. Claims about the moderating effects of age on the variables of interest should be supported by direct statistical analyses.
- ii) The behavioral data employed to identify event boundaries in the movie were obtained from a relatively small group of observers (n= 16, age range unspecified). It is unclear whether these participants were representative of those contributing to the fMRI data analyzed here, especially in respect of age. The authors should discuss the limitations imposed by these constraints.
- iii) Given that members of this research group have contributed to evidence indicating that the fMRI hemodynamic transfer function differs with age, it is somewhat surprising that the possibility that the reported age differences in neural state duration might be confounded by hemodynamic factors is not discussed.
- iv) The authors report the outcome of tests for linear relationships between state duration (and other variables) and age. Did they also test for non-linear relationships, as is customary in studies where age is treated as a continuous or near-continuous variable?

Reviewer #2

(Remarks to the Author)

Reviewer comments:

This study investigated how neural state segmentation differs with age by analyzing fMRI data from 577 adults (aged 18–88) who watched an 8-minute movie. Using a data-driven neural state segmentation method (Greedy State Boundary Search; GSBS), the authors found that neural states become significantly longer with age, particularly in the visual cortex and ventromedial prefrontal cortex, indicating reduced temporal differentiation. However, the alignment between neural states and perceived event boundaries remained stable across age groups, suggesting that coarse event segmentation is preserved. These findings suggest that while aging may lead to a blurring of neural states over time, the ability to segment major events remains intact.

This study is interesting and includes novel findings that make an important contribution to the fields of event cognition and neurocognitive aging. However, there are several areas of the paper that require further clarification. I enumerate these in my comments below:

Major comments:

1. The authors find that two regions (visual cortex and vmPFC) have longer neural states with increased age, but that a different set of regions tracks boundary overlap with behaviorally identified boundaries, which are not related to age. I did not find this to be problematic, to be clear, but I did feel that these results were not fully squared with one another. I was wondering if there might be network-membership differences between these sets of regions. For example, is there any evidence that these ROIs are in distinct functional networks (e.g., default versus frontoparietal)? If so, this could be a point worth raising in the Discussion.

2. One of the main dependent variables of interest is the duration of data-driven neural states. Does this relate to the number of neural states that was identified by the GSBS? Did areas (e.g., visual cortex and vmPFC) that demonstrated an increase in neural state duration with age also result in fewer neural states with age? It would be helpful if the authors included a summary figure or table of the total number of states per age group (maybe just for visual and vmPFC, since these were the ROIs that reliably slowed with with age). For example, the authors note in the Discussion: "This suggests that longer (fewer) neural states within the same period may contribute to older adults experiencing time as passing more quickly, which is in line with findings from previous studies (John & Lang, 2015)." This seemingly directly relates longer with fewer neural states. Perhaps I just missed it, but I was unable to find any direct reporting of the number of neural states here, with one exception on page 10 where the authors seem to mention the number of state changes for two example regions. Nonetheless, it is unclear if this is just from the actual GSBS results or from the data with simulated noise added:

"While with advancing age, the number of states can decline very steeply (e.g. from 41 to 25 states in the SOG), the simulations show a much smaller decrease (largest difference is from 41 to 37 in the SFG), suggesting that increased variability in the timing of state boundaries cannot explain the observed increase in neural state durations with age."

3. I wasn't sure I fully understood the absolute overlap metric, and I think more information would help with clarity. Does this measure take into consideration the total number of neural states found for a specific age group/individual? For example, let's say the younger adult group (age 18-42) had 10 neural states and all 9 of those state changes overlapped with a perceived/behaviorally identified event boundary compared to the older group (age 64-88), which only had 3 neural states and those two state changes overlapped with perceived/behaviorally identified event boundaries. How does this metric differentiate between these two scenarios? Would these both result in a similarly high absolute overlap metric, because in both cases there was 100% overlap between neural and behavioral boundaries? Or would the older adult group be down-weighted and get a lower absolute overlap value because they had less neural states to start with compared to the younger adult group?

Clarifying this aspect of the measure is important and makes a difference in how to interpret the findings. This relates to my difficulty in parsing the findings that there are longer neural states with age, yet preserved event boundary overlap. If the two young and the old age groups from the example above are treated the same way, then this makes more sense. In other words, in one case, older adults have fewer neural states but the few they do have are highly overlapping with behavioral boundaries. If this is not the case, then I suppose it might simply be that there are different sets of brain regions, so linking these two findings might not be possible. More unpacking of this in the discussion would be very helpful.

4. In the control analyses, ISS seems to be used as a proxy for signal-to-noise or as some composite measure that includes both inter-individual variability and noise. The authors state on page 8: "ISS declines strongly with age (Campbell et al., 2015; Geerligs and Campbell, 2018), and it affected both by the inter-individual variability of participants, as well as by their signal-to-noise levels." This is reasonable enough, but it seems parsimonious to simply use tSNR as an index of noise, as tSNR requires fairly few logical connections to arrive at being an index of signal quality. So, then, why not simply include tSNR as a covariate, or in addition to ISS? (I want to note that my comment is not meant to tell the authors to use tSNR instead of ISS, but rather to ask for a bit of justification of the current approach.)

5. In the Weaker neural state boundaries only partly explain the effect of age on state duration section of the Results, I wasn't totally clear on how the authors were computing neural state boundary strength. This seems to be defined as less distinct neural patterns between adjacent neural states (if I interpreted this correctly), but I couldn't find a description of how this was actually computed. Was it RSA? Or was this derived from the Time x Time correlation matrices?

Minor comments:

1. Page 3, line 110: I think the ideas about "dedifferentiation" can be unpacked a bit more here. It is not immediately clear how more blurry neural patterns would necessarily be related to longer neural states. Less distinct states, yes, but the connection to why the states would be longer per se could be clarified.

2. Page 3 line 117, first paragraph of results: The statement "34 groups of similar age" is confusing and might be a typo because it seems to contradict with the previous statement on line 105 that "34 age groups spanning the adults lifespan". Are they different age groups? Or small groups that have the same age ranges in them?

3. Page 4: It would be helpful to include the size of searchlight here (it is in the Methods on page 17 but would be nice to have this minor detail specified earlier for a streamlined interpretation of the Results).

4. Page 6: It would be helpful to clarify in this section that the perceived event boundaries were all from coarse events (as noted in the Methods). My first thought was that these event boundaries aren't necessarily at one granularity (exclusively coarse or fine levels), and are likely a mix of both. Clearly stating that they are coarse boundaries is important for clarifying this point.

5. Page 8: Definitions of what an event TR is vs. a non-event TR are needed. Also, could these more accurately be describe as “on behavioral boundary” vs. “off behavioral boundary”?

6. Figure 1: The maps seems to be thresholded at 9.9s, and voxels that had state durations smaller than 9.9s (e.g. the smallest neural state reported in the text was 4.9s) are excluded. This thresholding should be mentioned in the figure caption. This comment also applies to the other brain map figures as well if there was other thresholding applied to the other brain maps. Unless all the subsequent brain maps are based on the voxels in this first map. This should also be stated if this is the case.

7. Figure 3: I am not sure about what is being depicted here. What is being correlated? The BOLD signal at ever TR? What is a time x time correlation matrix?

8. Typo page 13 line 444: “This has been shown to shown to”

Reviewer #3

(Remarks to the Author)

This study assessed neural state duration across age during the viewing of a narrative movie, and how changes in neural states may relate to changes in event structure. The methodology is straightforward, and the analyses look to be conducted well. I don't have concerns about design, methodology, or analysis. I do have concerns on the interpretability of these results. What do we learn from them? What new information have we gained? We see that older adults tend to show longer activity states across the whole brain. However, what does that mean? We get some general speculations from the authors (e.g. inhibitory control differences), but how can that speculation, or others that were offered, explain whole-brain state changes? How does it relate to event representation? What are these networks "doing?" How do these results inform our understanding of the mechanisms important for event understanding? Ultimately, I see that older adults have longer brain states and phases, but how does one know what that means? In its current state, I do not recommend this article for publication in Communications Biology.

Version 1:

Reviewer comments:

Reviewer #1

(Remarks to the Author)

The authors have satisfactorily addressed the concerns raised in my review. No further comments.

Reviewer #2

(Remarks to the Author)

The authors have responded thoughtfully and thoroughly to my comments. I have no further critiques or suggestions at this time.

Reviewers' comments:

Reviewer #1 (Remarks to the Author):

This manuscript describes an extension of a secondary analysis of an existing dataset previously reported by the same group. Here, whole brain fMRI data that had been continuously acquired during the viewing of a short movie was analyzed to segment the fMRI timeseries into distinct 'states'. The key topics of interests were regional differences in the duration of different brain states, whether and where state transitions overlapped with a separate sample's judgments of event-transitions in the movie itself, and the effects of age on these variables. It was reported that state duration was positively correlated with age in regions demonstrating both short (visual cortex) and long (PFC) state durations, but that there was no effect of age on the extent that state transitions mapped onto reported event boundaries in the movie (these transitions were confined to regions where state duration was little affected by age). It is concluded that the age differences in state duration that were identified are a reflection of age-related 'temporal dedifferentiation', and that the event segmentation results are consistent with reports of null findings for the effects of age on event boundary judgments.

The results extend the authors' original report and will be of interest to those interested in age differences in brain dynamics and in perceptual and event representations. The paper would benefit from attention the following issues:

i) Some of the conclusions about age differences appear to be based on visual inspection rather than statistical analysis. For example, the assertion that the correlation matrices illustrated in figure 3 differ according to age does not seem to be supported statistically. Claims about the moderating effects of age on the variables of interest should be supported by direct statistical analyses.

We appreciate the reviewer's concerns, but think there may have been some misunderstanding. None of our claims are based on visual inspection alone. For instance, the matrices in Figure 3 were simply intended as visualizations to support the statistical results. We have attempted to make the statistical tests clearer throughout.

In the second section of the results (*Increase in neural state duration with age*) we look at the effect of age on median state duration. A Spearman's rank correlation coefficient was estimated per searchlight and FDR corrected across searchlights. Significant results are shown in figure 2. Whereas figure 1 is an unthresholded visualization, figure 2 shows where the effects of age are significant. From the results of the Spearman's rank correlation coefficient estimations, we selected the two searchlights with the highest correlation between age and duration. For those two searchlights, the time-by-time correlations are plotted in Figure 3 (now Fig 2C). Therefore, the conclusion of longer states with increasing age is based on the correlation and not the figure. Figure 2C also suggests weaker state boundaries and weaker within state correlations, which is statistically tested in the last section of the results (*Weaker neural state boundaries only partly explain the effect of age on state duration*, Figure 6 and S-XI).

To make the link between these different pieces of information clearer, we have combined figure 2 and 3 into one figure, added the correlations between age and state duration for the two selected searchlights (page 5), and reworded this section. It now reads:

"To visualize the effect of age on median state duration, we selected two searchlights in different clusters with the strongest effect of age, one in the superior occipital gyrus (SOG; $r_s = .86$) and one in the vmPFC ($r_s = .78$), and plotted time-by-time correlations for the youngest, middle, and oldest

groups (Figure 2B, i.e., groups 1, 17, and 34 in Supplementary Table S-I; see also Supplementary Figure S-I for the relation between age group and number of states). Figure 2B shows an example of how with increased age neural states become longer. It also shows how correlations within states become weaker with age. With additional analyses and simulations, we show that this cannot fully explain the increased state duration we see in older adults (see sections on simulations and boundary strength)."

We thoroughly reviewed the manuscript and explicitly included descriptions of the statistical tests and multiple testing corrections that had been performed but were previously not made explicit.

ii) The behavioral data employed to identify event boundaries in the movie were obtained from a relatively small group of observers (n= 16, age range unspecified). It is unclear whether these participants were representative of those contributing to the fMRI data analyzed here, especially in respect of age. The authors should discuss the limitations imposed by these constraints.

The reviewer raises a thoughtful point. The event boundaries are indeed based on a relatively small sample. However, as a recent study showed (Sasmita, K., & Swallow, K. M., 2023), stable segmentation agreement across subsamples for movie data can be achieved by samples between 10 and 16 people. The boundaries reported in Ben-Yakov and Henson (2018) have been used in multiple studies before, including Geerligs et al. (2022) to which our analyses are an extension. Age details were not reported in the original paper. We contacted the first author, who said they didn't keep a record of exact age but informed us that these were mainly younger adults. For this specific movie, Reagh et al. (2020) reported that there appeared to be no age differences in perceived event boundaries. Nevertheless, we have included this possible limitation in the manuscript on page 16:

"A potential limitation is that subjective event boundaries were defined using data from an independent sample of 16 younger adults (Ben-Yakov & Henson, 2018). However, Reagh et al. (2020) found no significant differences in event segmentation agreement between 14 older and 14 younger adults for the same movie. Additionally, recent work has demonstrated that stable event boundary agreement can be achieved with relatively small samples (Sasmita & Swallow, 2023). Finally, regions showing the strongest age-related effects in our data had limited overlap with those showing the greatest alignment between neural states and subjective boundaries. Together, these findings suggest that age-related differences in event boundary perception are unlikely to account for our results."

iii) Given that members of this research group have contributed to evidence indicating that the fMRI hemodynamic transfer function differs with age, it is somewhat surprising that the possibility that the reported age differences in neural state duration might be confounded by hemodynamic factors is not discussed.

In previous research on neural state segmentation, we investigated how differences in HRF shape may impact estimated neural state durations (Geerligs et al, 2022). There, we found that there were minimal effects of HRF shape on state durations. We have now explained that more clearly in the discussion section under *Validation of our results* (page 15-16):

"Previous studies have documented age-related changes in the hemodynamic response function; however, it is unlikely that the age-related increase of several seconds in neural state durations observed in our study can be explained by these changes, as they are typically less than one second in magnitude (West et al., 2019). For a more extensive discussion on limited effects of hemodynamic response delays on neural state duration estimates, see Geerligs et al., (2022)."

In the methods section, under *Image acquisition and preprocessing*, we explain why we did not apply hemodynamic response function deconvolution (page 18):

“We did not apply hemodynamic response function deconvolution to the data before detecting neural states. This is because deconvolution was found to be beneficial for detecting very brief states of 1 or 2 TRs, but unfortunately also reduces the distinctiveness of neural state boundaries due to lower signal-to-noise ratio in the data (Geerligs et al., 2022).”

iv) The authors report the outcome of tests for linear relationships between state duration (and other variables) and age. Did they also test for non-linear relationships, as is customary in studies where age is treated as a continuous or near-continuous variable?

We appreciate the reviewer’s insightful comment regarding the potential for non-linear age-related effects. In our main analyses, age was not treated as a continuous variable. Instead, we used Spearman’s rank correlation coefficients, which are well-suited to detecting monotonic relationships without assuming a specific functional form, and are appropriate given the use of age groups with a relatively limited number of data points (1 per group).

To more directly address the possibility of non-linear effects, we conducted supplementary analyses comparing linear, exponential, logarithmic, and quadratic polynomial models. These models were used to examine the relationship between age group and median neural state duration, and between continuous age and neural boundary strength. Model selection was based on the Akaike Information Criterion (AIC), which balances model fit and complexity.

For median state duration, a linear model provided the best fit, suggesting a predominantly linear relationship with age. For neural boundary strength, a quadratic model was superior, capturing a non-linear pattern across the lifespan. Specifically, boundary strength peaked in middle-aged adults, but older adults showed significantly lower strength than both middle-aged and younger adults. This indicates that although the relationship is non-linear, the overall trajectory reflects a decline in boundary strength from young to older age, with a temporary increase in midlife. These findings have been added to the supplementary materials and referred to in the main manuscript.

Results *Increase in neural state duration with age*, page 5:

“Supplementary analyses (SX-III) show that the association between age and state duration was mostly linear.”

Results *Weaker neural state boundaries only partly explain the effect of age on state duration*, page 11:

“Additionally, we tested for different relationships between age and state duration and strength (see supplementary analyses S-XIII) and found that whereas state duration linearly increases with age, boundary strength shows a quadratic relationship on top of a general decline.”

Supplementary materials Analyses S-XIII, page 38 and 39:

“Analyses S-XIII Model comparison for age relationships with state duration and boundary strength

In the main manuscript, we examined all relationships with age using Spearman’s rank correlation coefficients. This non-parametric method is advantageous as it detects monotonic trends without assuming a specific functional form and accommodates categorical variables, in our case age groups.

Here, we present supplementary analyses aimed at identifying the optimal functional form of the age–neural state relationship. Specifically, we assessed the relationship between age groups and median state duration, and between continuous age and neural boundary strength. We compared four models—linear, quadratic, exponential, and logarithmic—using the Akaike Information Criterion (AIC) for model selection. For each searchlight, all four models were fit using curve fitting, and the model with the lowest AIC was selected, balancing goodness-of-fit with model complexity.

For the relationship between age group and median state duration, linear models provided the best fit in 60.5% of searchlights, followed by quadratic (21.8%), exponential (15.6%), and logarithmic (2.1%) models. This suggests that the relationship between age and state duration is predominantly linear across the brain.

In contrast, for the relationship between continuous age and boundary strength, quadratic models were optimal in 72.8% of searchlights, followed by linear (25.8%), logarithmic (0.9%), and exponential (0.5%) models. The AIC differences between quadratic and linear models indicate nonlinear age-related effects on boundary strength.

In line with the results in our main manuscript, these additional analyses suggest that aging has a different effect on neural state duration than on boundary strength. Whereas state duration increases linearly in large parts of the brain with age, boundary strength follows a quadratic shape with the highest strength in the middle of adulthood but still a lower tail in older age compared to younger age. To illustrate this, the figure below shows the relationship between age and boundary strength for the six searchlights with the biggest AIC difference between a linear and quadratic fit.”

“Figure S-XIII Age - boundary strength relationships for six searchlights with the largest AIC

differences favoring a quadratic over a linear fit. Each panel shows individual data points for all searchlights, with linear (blue) and quadratic (red) trend lines overlaid.”

Reviewer #2 (Remarks to the Author):

Reviewer comments:

This study investigated how neural state segmentation differs with age by analyzing fMRI data from 577 adults (aged 18–88) who watched an 8-minute movie. Using a data-driven neural state segmentation method (Greedy State Boundary Search; GSBS), the authors found that neural states become significantly longer with age, particularly in the visual cortex and ventromedial prefrontal cortex, indicating reduced temporal differentiation. However, the alignment between neural states and perceived event boundaries remained stable across age groups, suggesting that coarse event segmentation is preserved. These findings suggest that while aging may lead to a blurring of neural states over time, the ability to segment major events remains intact.

This study is interesting and includes novel findings that make an important contribution to the fields of event cognition and neurocognitive aging. However, there are several areas of the paper that require further clarification. I enumerate these in my comments below:

Major comments:

1. The authors find that two regions (visual cortex and vmPFC) have longer neural states with increased age, but that a different set of regions tracks boundary overlap with behaviorally identified boundaries, which are not related to age. I did not find this to be problematic, to be clear, but I did feel that these results were not fully squared with one another. I was wondering if there might be network-membership differences between these sets of regions. For example, is there any evidence that these ROIs are in distinct functional networks (e.g., default versus frontoparietal)? If so, this could be a point worth raising in the Discussion.

We thank the reviewer for this excellent suggestion. The finding that the regions in which neural state duration is most affected by age differs from the areas with the highest overlap between neural states and perceived event boundaries is indeed a key result of our paper and aligns with studies suggesting no difference in coarse event segmentation with age.

As suggested, we examined which networks overlap with areas that show the strongest effects of age on neural state duration versus those with the highest overlap between neural states and event boundaries. To provide a comprehensive analysis, we used the Schaefer 7-network 200 parcellation and compared strength of effects per network.

We have added these results to the supplementary materials and included a section on this in the discussion (page 15):

“Network-level analyses revealed distinct patterns for the two processes examined (see supplementary Figure S-XII). While age-related increases in neural state duration were most pronounced in the visual network, the correspondence between perceived event boundaries and neural state boundaries was highest in attention-related networks (primarily salience/ventral attention). This network-level distinction suggests that aging affects neural state dynamics primarily in visual processing regions (fitting with the above-described findings on inhibitory control), while event boundary detection might rely on networks involved in salience processing and detection of relevant information from the environment (Uddin et al., 2019).”

Supplementary materials S-XII (page 38):

“Figure S-XII Network involvement in the correlation between age group and median neural state duration on the left and overlap between perceived event boundaries and neural states on the right.

Our results suggest that a different set brain areas is involved in event boundary detection than those most affected by age. Here we investigated network-involvement of those effects using the Schaefer 7-network 200 parcellation by comparing the strength of effects per network. Age-related increases in neural state duration were most pronounced in the visual network, whilst the correspondence between perceived event boundaries and neural state boundaries was highest in the salience/ventral attention network.”

2. One of the main dependent variables of interest is the duration of data-driven neural states. Does this relate to the number of neural states that was identified by the GSBS? Did areas (e.g., visual cortex and vmPFC) that demonstrated an increase in neural state duration with age also result in fewer neural states with age? It would be helpful if the authors included a summary figure or table of the total number of states per age group (maybe just for visual and vmPFC, since these were the ROIs that reliably slowed with age). For example, the authors note in the Discussion: “This suggests that longer (fewer) neural states within the same period may contribute to older adults experiencing time as passing more quickly, which is in line with findings from previous studies (John & Lang, 2015).” This seemingly directly relates longer with fewer neural states. Perhaps I just missed it, but I was unable to find any direct reporting of the number of neural states here, with one exception on page 10 where the authors seem to mention the number of state changes for two example regions. Nonetheless, it is unclear if this is just from the actual GSBS results or from the data with simulated noise added:

“While with advancing age, the number of states can decline very steeply (e.g. from 41 to 25 states in the SOG), the simulations show a much smaller decrease (largest difference is from 41 to 37 in the SFG), suggesting that increased variability in the timing of state boundaries cannot explain the observed increase in neural state durations with age.”

Thank you for alerting us that this was unclear. You are right – longer states means fewer states. To hopefully make this clearer, we have made a number of changes in the manuscript and the supplementary materials.

In the first paragraph of the results section (page 4), we added the following text:

“This optimal number of states is inversely related to the neural state duration as the duration of the movie stimulus was constant.”

In the section *Increase in neural state duration with age* of the Results (page 5) we added a direct comparison between the youngest and oldest groups in state duration:

“All 2816 significant searchlights (FDR-corrected) showed positive correlations between age and state duration. Direct comparison between the youngest and oldest groups confirmed this pattern, with 93.9% of significant regions showing longer durations in older adults (mean difference = 8.36 seconds, SD = 5.02, range -9.88 – 32.11; Supplementary Figure S-IA). In the remaining 6.1% the difference between the youngest and oldest group was either neutral or negative but across groups all significant correlations were positive.”

Supplementary Figure S-I (page 31):

“Figure S-I A) Comparison of median state duration between the youngest group (left) and oldest group (right) for all 2816 significant searchlights that showed an effect of age. Bars show means ± SEM, dots show individual searchlights. Mean $\Delta = 8.36$ seconds, $p < .001$ B) The effect of age group on number of neural states for the vmPFC and SOG.

Figure S-I shows in panel A for all searchlights with a significant effect of age (after FDR correction) how the median state duration changed with age from the youngest group to the oldest group. The mean change is an increase of 8.36 seconds in state duration (SD = 5.02, range -9.88 – 32.11). 93.9% of significant regions showed longer durations in older adults, in the remaining 6.1% the difference between the youngest and oldest group was either neutral (3.1%) or negative (3.0%) but across all groups all significant correlations were positive. How the number of states can fluctuate across age despite a negative correlation is illustrated in panel B. This gives an example of how the number of neural states changes with increasing age for two searchlights that showed the strongest effect of age. The number of neural states is the inverse of the duration of neural states as both are based on the neural state boundaries estimated by GSBS. In the main manuscript, the median duration

between the neural state boundaries is used as measure of interest. Panel B shows the count of the neural states for illustrative purposes.”

3. I wasn't sure I fully understood the absolute overlap metric, and I think more information would help with clarity. Does this measure take into consideration the total number of neural states found for a specific age group/individual? For example, let's say the younger adult group (age 18-42) had 10 neural states and all 9 of those state changes overlapped with a perceived/behaviorally identified event boundary compared to the older group (age 64-88), which only had 3 neural states and those two state changes overlapped with perceived/behaviorally identified event boundaries. How does this metric differentiate between these two scenarios? Would these both result in a similarly high absolute overlap metric, because in both cases there was 100% overlap between neural and behavioral boundaries? Or would the older adult group be down-weighted and get a lower absolute overlap value because they had less neural states to start with compared to the younger adult group?

Clarifying this aspect of the measure is important and makes a difference in how to interpret the findings. This relates to my difficulty in parsing the findings that there are longer neural states with age, yet preserved event boundary overlap. If the two young and the old age groups from the example above are treated the same way, then this makes more sense. In other words, in one case, older adults have fewer neural states but the few they do have are highly overlapping with behavioral boundaries. If this is not the case, then I suppose it might simply be that there are different sets of brain regions, so linking these two findings might not be possible. More unpacking of this in the discussion would be very helpful.

Thank you for pointing out that this was unclear. To better explain the absolute overlap measure, we have added additional explanations (page 20) with an example to help the reader get a better intuition for this metric.

“To assess the overlap in neural state boundaries and perceived event boundaries, we used the absolute overlap metric described by Geerligs et al. (2022). The absolute overlap computes the total number of event and neural states boundaries that overlap and subsequently scales this with respect to the total number of boundaries as the maximum value and the expected number of overlapping boundaries as the minimum value. A measure of one indicates that all neural state boundaries (per searchlight, per age group) align with an event boundary (same for all searchlights and age groups), zero indicates an overlap as expected by chance. For example, imagine there are 20 neural states boundaries, 10 event boundaries and 100 time points, of which 8 overlap. In this case the expected overlap would be $20/100 * 10/100 * 100 = 2$. The absolute overlap would be $(8-2)/(20-2) = 0.334$. In practice, the scaling with respect to the expected overlap provides an appropriate baseline, but it does not affect the observed differences between regions or groups. This means that all regional and age-related variability reported in the paper is identical compared to simply looking at the percentage of all neural state boundaries that overlap with an event boundary. Therefore, if one group has 20 boundaries, of which 10 overlap with event boundaries and the other group has 30 boundaries, of which 15 overlap with event boundaries, these groups will have a nearly identical absolute overlap.”

This is now also mentioned in the discussion (page 15):

“It is important to note here that in quantifying this correspondence, we looked at the proportion of neural state boundaries that overlap with an event boundary. This means that even if participants differ in their total number of neural state boundaries, they can still show similar proportions of overlap. Our findings therefore suggest that, among the neural state boundaries that do occur, their

tendency to align with event boundaries is similar across younger and older participants. However, this does not imply that the actual number of event boundaries that align with neural state boundaries is the same across age groups.”

4. In the control analyses, ISS seems to be used as a proxy for signal-to-noise or as some composite measure that includes both inter-individual variability and noise. The authors state on page 8: “ISS declines strongly with age (Campbell et al., 2015; Geerligs and Campbell, 2018), and it is affected both by the inter-individual variability of participants, as well as by their signal-to-noise levels.” This is reasonable enough, but it seems parsimonious to simply use tSNR as an index of noise, as tSNR requires fairly few logical connections to arrive at being an index of signal quality. So, then, why not simply include tSNR as a covariate, or in addition to ISS? (I want to note that my comment is not meant to tell the authors to use tSNR instead of ISS, but rather to ask for a bit of justification of the current approach.)

We appreciate the reviewer’s insightful comment regarding the distinction between tSNR and ISS. We agree that tSNR is a well-established metric for assessing low-level signal stability. However, our choice to use ISS was motivated by its capacity to capture not only signal noise (as the intersubject synchrony measure should decline with the presence of noise) but also time-locked synchronization across participants. Therefore, ISS is sensitive to both noise and inter-individual variability in neural processing over time. We have added this clarification to our methods section (page 21):

“Unlike low-level signal quality metrics such as temporal signal-to-noise ratio (tSNR), ISS captures both noise-related reductions in signal stability as well as inter-individual differences in time-locked neural processing.”

5. In the Weaker neural state boundaries only partly explain the effect of age on state duration section of the Results, I wasn’t totally clear on how the authors were computing neural state boundary strength. This seems to be defined as less distinct neural patterns between adjacent neural states (if I interpreted this correctly), but I couldn’t find a description of how this was actually computed. Was it RSA? Or was this derived from the Time x Time correlation matrices?

Boundary strength is indeed based on how distinct neural patterns are between adjacent states. This is calculated with a Pearson correlation between the average neural activity patterns of consecutive states and is part of the GSBS output. In the *material and method section* we describe it as follows:

“To compute individual boundary strength using GSBS, the first step was to determine a shared set of boundaries by applying neural state segmentation to the entire group of participants. Next, we investigated the strength of these boundaries within each individual. Here, boundary strength is defined as 1 minus the Pearson correlation of the average neural activity patterns of the two neural states surrounding each boundary. Thus, a higher strength implies more dissimilarity between two consecutive neural states.”

For clarity we have added this to the results section (page 11):

“Theoretically, neural state boundary strengths can vary between just above 0 and 2 (1 minus the correlation between the neural activity patterns of two adjacent states), with higher values indicating that two successive states are more dissimilar.”

Minor comments:

1. Page 3, line 110: I think the ideas about “dedifferentiation” can be unpacked a bit more here. It is not immediately clear how more blurry neural patterns would necessarily be related to longer neural

states. Less distinct states, yes, but the connection to why the states would be longer per se could be clarified.

To make this clearer, we rephrased the last part of this paragraph (page 3):

“Based on prior findings of spatial dedifferentiation in the aging brain, which suggest reduced distinctiveness of neural representations for different objects and categories (e.g., Koen & Rugg, 2019; Koen et al., 2020), we hypothesized that aging would be associated with less distinct neural states over time. Additionally, drawing on evidence of reduced suppression of previously attended information in older adults (e.g., Hamm & Hasher, 1992; Scullin et al., 2011; Weeks et al., 2020; Campbell, Lustig, & Hasher, 2020), we expected aging to be associated with longer-lasting neural states.”

2. Page 3 line 117, first paragraph of results: The statement “34 groups of similar age” is confusing and might be a typo because it seems to contradict with the previous statement on line 105 that “34 age groups spanning the adults lifespan”. Are they different age groups? Or small groups that have the same age ranges in them?

Thank you for calling our attention to this. We rephrase this to (page 3):

“Five hundred seventy-seven adults (age range 18-88) were split into 34 groups, each consisting of participants within a specific age bracket (e.g., 18-23, 23-25, ..., 83-88), to assess effects of age on neural state boundaries identified within each group.”

3. Page 4: It would be helpful to include the size of searchlight here (it is in the Methods on page 17 but would be nice to have this minor detail specified earlier for a streamlined interpretation of the Results).

Good suggestion. We’ve added this at the top of our results (page 3-4):

“Data were hyperaligned (Guntupalli et al., 2016) per group before applying GSBS to 5402 searchlights (average searchlight size of 97 voxels) covering the entire cortex.”

4. Page 6: It would be helpful to clarify in this section that the perceived event boundaries were all from coarse events (as noted in the Methods). My first thought was that these event boundaries aren't necessarily at one granularity (exclusively coarse or fine levels), and are likely a mix of both. Clearly stating that they are coarse boundaries is important for clarifying this point.

We have added the instructions on segmentation to the results section (page 7):

“We used the absolute overlap metric described by Geerligs et al. (2022) to assess the overlap between neural state boundaries and perceived event boundaries. These perceived event boundaries were based on agreement of when one meaningful event ended and the next began in a separate group of 16 subjects (Ben-Yakov and Henson, 2018).”

5. Page 8: Definitions of what an event TR is vs. a non-event TR are needed. Also, could these more accurately be describe as “on behavioral boundary” vs. “off behavioral boundary”?

Thank you, we have adopted this wording in the paragraph *Preserved overlap between neural states and events with age* and caption of figure 4 (page 8):

“To investigate if alignment with event boundaries made a difference for the effect of age on neural boundary occurrence, we examined the occurrence of neural states separately for TRs that overlap with event boundaries (on event boundary) and those that do not (off event boundary) across all searchlights. In both on event boundary and off event boundary TRs we see an overall decrease in boundary occurrence with age (i.e., longer states), however this effect is stronger for off event boundary TRs (mean correlation across all searchlights $r_s = -0.36$) than on event TRs ($r_s = -0.21$; Figure 4B, see also Supplementary Figure S-II). This explains why we observe longer neural states with increasing age without a decrease in overlap between event and neural state boundaries. Collectively, these findings suggests that the relationship between neural states and perceived event boundaries remains largely similar with age.”

6. Figure 1: The maps seems to be thresholded at 9.9s, and voxels that had state durations smaller than 9.9s (e.g. the smallest neural state reported in the text was 4.9s) are excluded. This thresholding should be mentioned in the figure caption. This comment also applies to the other brain map figures as well if there was other thresholding applied to the other brain maps. Unless all the subsequent brain maps are based on the voxels in this first map. This should also be stated if this is the case.

We thank the reviewer for this helpful observation. To clarify, the maps are not thresholded in terms of voxel inclusion or exclusion; rather, the color scale was limited for visualization purposes. All voxels are included in the maps, and the full range of data is represented. However, for clarity and to optimize visual contrast, the color scale was restricted to durations between 4 and 9 TRs, which encompasses the majority of searchlight values. Voxels with durations outside this range are still shown but are displayed using the color corresponding to the nearest bound.

To prevent confusion, we have revised the figure caption (Figure 1, pages 4–5) to clarify this point. Specifically, we now state:

“The full range of median state durations across searchlights across 34 age groups was 2 to 18 TRs, however, averaging over groups resulted in a range of durations from 2.6 to 10.8 TRs. For visualization purposes, the color scale is limited to values between 4 and 9 TRs, as most searchlights fell within this range. Voxels with values outside this range are displayed using the color corresponding to the nearest bound (i.e., values <4 TRs are shown as 4 TRs; values >9 TRs are shown as 9 TRs).”

We have also specified the range of values in other figure captions.

7. Figure 3: I am not sure about what is being depicted here. What is being correlated? The BOLD signal at ever TR? What is a time x time correlation matrix?

Thank you, we have changed the figure caption (now Figure 2B, page 6) to explain this:

“B) Locations of the searchlights that showed the highest correlation between age and median state duration on the left, followed by time-by-time correlation matrices (correlation based on the averaged brain activity time course in a searchlight for each TR) for the youngest, middle, and oldest group (left to right). White boxes represent the neural state boundaries detected by GSBS.”

8. Typo page 13 line 444: "This has been shown to shown to"

Thank you, we have deleted the repetition.

Reviewer #3 (Remarks to the Author):

This study assessed neural state duration across age during the viewing of a narrative movie, and how changes in neural states may relate to changes in event structure. The methodology is straightforward, and the analyses look to be conducted well. I don't have concerns about design, methodology, or analysis. I do have concerns on the interpretability of these results. What do we learn from them? What new information have we gained? We see that older adults tend to show longer activity states across the whole brain. However, what does that mean? We get some general speculations from the authors (e.g. inhibitory control differences), but how can that speculation, or others that were offered, explain whole-brain state changes? How does it relate to event representation? What are these networks "doing?" How do these results inform our understanding of the mechanisms important for event understanding? Ultimately, I see that older adults have longer brain states and phases, but how does one know what that means? In its current state, I do not recommend this article for publication in Communications Biology.

We appreciate the reviewer's recognition that our design, methodology, and analyses are sound. We are also grateful for the opportunity to clarify the interpretability and significance of our findings. Below, we address the reviewer's concerns point by point and have revised the manuscript accordingly to emphasize the relevance of our findings to theories of event segmentation, underlying neural dynamics, and aging.

1. What do we learn from [the results]? What new information have we gained?

Our study provides novel insights into how aging affects the temporal dynamics of brain activity during naturalistic event processing. Specifically, we identify which aspects of neural dynamics are altered with age and which remain preserved. Our key findings are:

1) The temporal hierarchy of neural state durations across the cortex (i.e., shorter neural states in sensory areas, longer in high-level regions) remains stable with age, suggesting preserved large-scale organizational principles of neural processing during naturalistic viewing.

2) We are the first to show (to our knowledge) that neural states become significantly longer with age, particularly in the vmPFC and visual cortex. We suggest that this finding might be related to previous findings that older adults maintain access to previously attended information that is no longer relevant to the task at hand.

3) Older adults show weaker state boundary strength (i.e., consecutive neural states are less dissimilar), aligning with prior evidence of neural dedifferentiation and reduced representational specificity in aging.

4) Despite the above changes, the alignment between neural state transitions and perceived event boundaries is maintained across age groups. This preservation suggests that at least some aspects of event segmentation are stable throughout the adult lifespan, which could be explained by the difference between areas affected by age and those with the strongest overlap between perceived event boundaries and neural state boundaries.

2. We see that older adults tend to show longer activity states across the whole brain. However, what does that mean? We get some general speculations from the authors (e.g. inhibitory control differences), but how can that speculation, or others that were offered, explain whole-brain state changes?

Respectfully, this is not strictly true. The manuscript shows that only a *limited* set of brain regions, mainly the vmPFC and visual cortex, show a statistically significant effect of age on neural state duration (section 2 of the results *Increase in neural state duration with age* and figure 2). The section of the discussion in which we discuss the relation between age and neural state changes states in the first paragraph where we find these changes (page 13):

“Longer and less distinct neural states with age

Our results show a strong effect of age on the duration of neural states, especially in the visual cortex and vmPFC. Neural state duration in the visual cortex has been related to perceptual processing of visual changes in a movie (Oetinger et al., 2024) which changes with age (Faubert, 2002). One of the processes the vmPFC is involved in is future simulation and integration of schematic knowledge (Benoit et al., 2014). The medial PFC is also a key node in the default mode network, which in aging has been related to reduced suppression of representations and increased use of associations and knowledge of the world (Spreng and Turner, 2019).”

In our revised manuscript we make it clearer that our points about reduced inhibitory control and dedifferentiation with age relate to our finding of altered neural state dynamics in visual processing regions where we report strong effects of age both on duration and boundary strength (page 13).

“Another possible mechanism is reduced inhibitory control with age (Hasher & Zacks, 1988). Previous work suggests that older adults maintain access to previously attended information that is no longer relevant to the task at hand and that this is related to perceptual processing areas (e.g., Hamm & Hasher, 1992; Scullin et al., 2011; Weeks et al., 2020; for a review, see Campbell, Lustig, & Hasher, 2020). If older adults still have the previous event in mind as the next event starts, this could lead to some missed state boundaries and/or weaker boundary strengths with age. Indeed, we observed a decrease in number of boundaries and boundary strength with age in different regions of the visual cortex; that is, older adults showed a longer state duration and less change in the pattern of neural activity between successive states (Figure 2 and 6). [...] Therefore, the temporal dedifferentiation we see as expressed in weaker boundaries between states or fewer distinct states in the visual cortex could be a direct consequence of spatial dedifferentiation.”

3. What are these networks "doing?"

In the previous version of the manuscript, we did not relate our results to any specific predefined networks and only investigated changes on the level of searchlights. However, in our revised manuscript we have added a supplementary analysis in which we relate the strongest significant findings of age on state duration and of overlap between perceived event boundaries and neural state boundaries to larger functional networks (as requested by R2). This can be found on page 15:

“Network-level analyses revealed distinct patterns for the two age-related processes examined (see supplementary Figure S-XII). While age-related increases in neural state duration were most pronounced in the visual network, the correspondence between perceived event boundaries and neural state boundaries was highest in attention-related networks (primarily salience/ventral attention). This network-level dissociation suggests that aging affects neural state dynamics primarily in visual processing regions (fitting with the above-described findings on inhibitory

control), while event boundary detection might rely on networks involved in salience processing and detection of relevant information from the environment (Uddin et al., 2019).”

In conclusion, while our paper focused on local neural state dynamics, our findings offer valuable insights that can be considered in the context of whole-brain networks.

4. How does it relate to event representation? How do these results inform our understanding of the mechanisms important for event understanding?

We do not exactly know how duration of neural states relates to perception or memory but what we do know is that there is a nested hierarchy of duration in neural states across different brain regions. High-level regions have neural state boundaries that correspond most closely to perceived event boundaries (Baldasano et al., 2017; Geerligs et al., 2022). We extend previous findings by showing that this cortical organization remains stable across the adult life span. Baldasano et al. (2017) have shown that segmentation at the neural level is relevant for memory storage and retrieval of the current event model. Additionally, a recent study from our lab has shown that greater distinction between successive neural states relates to better episodic memory.

We have added the following sentences to the discussion to stress why changes in neural state duration might matter (page 14):

“Furthermore, Baldassano et al. (2017) have associated neural state boundaries to storing of the current event model in memory, and we recently showed that greater distinction between successive neural states relates to better episodic memory in both younger and older adults (Henderson et al., 2025). Therefore, longer and less distinct neural states with age might influence the formation of memories for events in the movie. This hypothesis needs to be tested in future research.”

The manuscript also discusses other reasons why our findings of longer and weaker states with age are interesting in the context of event cognition. Specifically, it suggests that these findings might be related to reduced perceptual inhibition and could indicate a blurring of events over time. This means that subsequent events have a tendency to merge and distinctions between events in time becomes less precise. Additionally, the manuscript discusses how time perception is linked to accumulation of salient stimuli changes. With fewer neural state changes, a given amount of time would be perceived to last shorter with age.

The finding that in areas where perceived events and neural states overlap strongest is unaffected by age is important as it suggests that coarse event segmentation remains intact with age despite changes in neural state dynamics in other areas. We suggest that future research can further investigate those changes (page 16).

“Future studies should determine if the boundaries that disappear with age are those that align with specific types of events. These could be event boundaries that have lower interrater agreement across participants on an event segmentation task or changes in specific aspects of the stimulus like spatial location or goals of the actor.”

In the limitation section (page 16) we acknowledge the constraints to interpreting our work more clearly:

“More broadly, much is still unknown about what neural states represent and how individual differences in neural states might shape differences in event segmentation and related cognitive

processes like perception, action and memory. This makes it difficult to make strong claims about how the age-differences we observed here affect the processing of events. More generally though, our approach offers a way to investigate the changes in segmentation on the neural level, which are a strong neural correlate of event segmentation, without altering the mental state of the participant by instructing them to segment the movie into meaningful units. As such, it offers a stepping stone for follow-up research to identify exactly how the temporal dedifferentiation of neural states affects how events are represented and memorized.”

5. Ultimately, I see that older adults have longer brain states and phases, but how does one know what that means?

We hope that by answering the questions above in this rebuttal and where possible in the manuscript, we have demonstrated the relevance and meaning of our findings.